# Solving Sparse & High-Dimensional-Output Regression via Compression

**Renyuan Li**
Department of Industrial Systems Engineering & Management
National University of Singapore
renyuan.li@u.nus.edu

**Zhehui Chen**
Google
zhehuichen@google.com

**Guanyi Wang**
Department of Industrial Systems Engineering & Management
National University of Singapore
guanyi.w@nus.edu.sg

## Abstract

Multi-Output Regression (MOR) has been widely used in scientific data analysis for decision-making. Unlike traditional regression models, MOR aims to simultaneously predict multiple real-valued outputs given an input. However, the increasing dimensionality of the outputs poses significant challenges regarding interpretability and computational scalability for modern MOR applications. As a first step to address these challenges, this paper proposes a Sparse & High-dimensional-Output REgression (SHORE) model by incorporating additional sparsity requirements to resolve the output interpretability, and then designs a computationally efficient two-stage optimization framework capable of solving SHORE with provable accuracy via compression on outputs. Theoretically, we show that the proposed framework is computationally scalable while maintaining the same order of training loss and prediction loss before-and-after compression under arbitrary or relatively weak sample set conditions. Empirically, numerical results further validate the theoretical findings, showcasing the efficiency and accuracy of the proposed framework.

## 1   Introduction

Multi-Output Regression (MOR) problem [8, 44] is a preponderant tool for factor prediction and decision-making in modern data analysis. Compared with traditional regression models that focus on a scalar output for each sample, MOR aims to predict multiple outputs $\boldsymbol{y} \in \mathbb{R}^K$ *simultaneously* based on a given input $\boldsymbol{x} \in \mathbb{R}^d$, i.e.,

$$\boldsymbol{y} := \arg\min_{\boldsymbol{u} \in \mathcal{Y}} \texttt{dist}(\boldsymbol{u}, \widehat{g}(\boldsymbol{x})) \quad \text{with} \quad \widehat{g} := \arg\min_{g \in \mathcal{G}} \frac{1}{n} \sum_{i=1}^{n} \ell(\boldsymbol{y}^i, g(\boldsymbol{x}^i)) \ ,$$

where we use $\{(\boldsymbol{x}^i, \boldsymbol{y}^i)\}_{i=1}^n$ to denote its given sample set with $\boldsymbol{x}^i \in \mathbb{R}^d$ $i$-th input feature vector and $\boldsymbol{y}^i \in \mathbb{R}^K$ corresponding output vector, define $\ell : \mathbb{R}^K \times \mathbb{R}^K \to \mathbb{R}$ as the loss function, $\texttt{dist} : \mathbb{R}^K \times \mathbb{R}^K \to \mathbb{R}$ as some prediction/distance metric, $\mathcal{Y}$ as the structure/constraint set for multiple outputs, and $\mathcal{G}$ as the candidate set for predicting model $g : \mathbb{R}^d \to \mathbb{R}^K$. Hence, MOR and its variants have been used for numerous regression tasks with structure requirements on multi-dimensional outputs arising from real applications, such as simultaneous estimation of biophysical parameters from remote sensing images [40], channel estimation through the prediction of several received signals [35], the grounding (e.g., factuality check [16]) in the Large Language Model (LLM, [34, 11]) era, to name but a few.

38th Conference on Neural Information Processing Systems (NeurIPS 2024).

In this paper, we are interested in the interpretability issue of high-dimensional outputs obtained from modern MOR tasks. One typical example is raised from algorithmic trading. In particular, in algorithmic trading, MOR helps to construct the portfolio [31] from a large number of financial instruments (e.g., different stocks, futures, options, equities, etc [19]) based on given historical market and alternative data [22]. To be concise, a high-dimensional output in this example could be viewed as a "decision", where every component denotes the investment for the corresponding financial instruments. Thus, other than accuracy, quantitative researchers prefer outputs with only a few instruments to enhance interpretability for the underlying decision-making reasons, which naturally introduces a sparse output condition. Similar scenarios apply to other applications, including offline reinforcement learning in robotics[33], discovering genetic variations based on genetic markers[25].

As a result, the dramatic growth in output dimensions gives rise to two significant challenges: **1**. High-dimensional-output impedes human interpretation for decision-making; **2**. Approaches with better computational scalability are desired for training & predicting MOR. Upon these challenges, a conceptual question that motivates this research is:

*How to design a framework that predicts output with enhanced interpretability, better computational scalability, and provable accuracy under a modern high-dimensional-output setting?*

Generally speaking, this paper provides an affirmative answer as a first step to the above question. Before presenting the main contributions, let us first introduce the model that will be studied in this paper. Unlike the classical MOR model, we further assume that given outputs are of high-dimensional (i.e., $d \ll K$), and to address the interpretability issue, these outputs have at most $s$ non-zero components, i.e., $\|y^i\|_0 \leq s$, for all $i \in [n]$ with some pre-determined sparsity-level $s(\ll K)$. Based on such given samples, this paper proposes the (uncompressed) *Sparse & High-dimensional-Output REgression (SHORE)* model that aims to predict an interpretable high-dimensional output $y$ (i.e., $s$-sparse in this paper) via any input feature vector $x$. In particular, to be concise and still capture the essential relationship, the proposed (uncompressed) SHORE model predicts $y$ from $x$ under a linear model, i.e., $y = \arg\min_{\|y\|_0 \leq s} \texttt{dist}(y, \widehat{Z}x)$ for some distance metric (see Section 3.1, prediction stage) and the linear regression $\widehat{Z}$ is obtained by solving the following linear regression problem:

$$\widehat{Z} := \underset{Z \in \mathbb{R}^{K \times d}}{\arg\min}\, \widehat{\mathcal{L}}_n(Z) := \frac{1}{n}\|Y - ZX\|_F^2, \tag{1}$$

where $X := (x^1 \mid \cdots \mid x^n) \in \mathbb{R}^{d \times n}$ is the input matrix and $Y := (y^1 \mid \cdots \mid y^n) \in \mathbb{R}^{K \times n}$ is the corresponding *column-sparse* output matrix.

## 1.1 Contributions and Paper Organization

This paper makes the following three main contributions:

**1**. We propose a two-stage computationally efficient framework for solving SHORE model. Specifically, the first training stage offers a computationally scalable reformulation on solving SHORE through compression in the output space. The second prediction stage then predicts high-dimensional outputs from a given input by solving a specific sparsity-constrained minimization problem via an efficient iterative algorithm.

**2**. We show that for arbitrarily given samples, the training loss in the first stage with compression is bounded by a $1 + \delta$ multiplicative ratio of the training loss for the original one (1) with some positive constant $\delta$. Additionally, the proposed iterative algorithm in the second stage exhibits global geometric convergence within a neighborhood of the ground-truth output, with a radius proportional to the given sample's optimal training loss. Furthermore, if all samples are drawn from a light-tailed distribution, the generalization error bound and sample complexity remain in the same order for SHORE with output compression. This finding indicates that the proposed framework achieves improved computational efficiency while maintaining the same order of generalization error bounds statistically.

**3**. We conduct rich numerical experiments that validate the theoretical findings and demonstrate the efficiency and accuracy of the proposed framework on both synthetic and real-world datasets.

In summary, this paper studies the SHORE model through computational and statistical lenses and provides a computationally scalable framework with provable accuracy.

The paper is organized as follows: Section 2 reviews related literature; Section 3 presents our proposed framework and provides theoretical results on sample complexity and generalization error bounds; Section 4 compares the proposed method with existing baselines in a suite of numerical experiments on both synthetic and real instances. Concluding remarks are given in Section 5.

**Notation.** Given a positive integer $n$, we denote $[n] := \{1, \ldots, n\}$. We use lowercase letters $a$ as scalars and bold lowercase letters $\boldsymbol{a}$ as vectors, where $\boldsymbol{a}_i$ is its $i$-th component with $i \in [d]$, and bold upper case letters $\boldsymbol{A}$ as matrices. Without specific description, for a $m$-by-$n$ matrix $\boldsymbol{A}$, we denote $\boldsymbol{A}_{i,j}$ as its $(i, j)$-th component, $\boldsymbol{A}_{i,:}^\top$ as its $i$-th row, $\boldsymbol{A}_{:,j}$ as its $j$-th column. For a symmetric square matrix $\boldsymbol{A}$, we denote $\lambda_{\max}(\boldsymbol{A}), \lambda_{\min}(\boldsymbol{A})$ and $\lambda_i(\boldsymbol{A})$ as its maximum, minimum and $i$-th largest eigenvalue, respectively. We denote $\|\boldsymbol{a}\|_1, \|\boldsymbol{a}\|_2, \|\boldsymbol{a}\|_\infty, \|\boldsymbol{A}\|_F, \|\boldsymbol{A}\|_{\mathrm{op}}$ as the $\ell_1, \ell_2, \ell_\infty$-norm of a vector $\boldsymbol{a}$, the Frobenius norm and the operator norm of a matrix $\boldsymbol{A}$, respectively. We denote $\mathbb{I}(\cdot)$ as the indicator function, $\|\boldsymbol{a}\|_0 := \sum_{i=1}^d \mathbb{I}(\boldsymbol{a}_i \neq 0)$ as the $\ell_0$-norm (i.e., the total number of nonzero components), $\mathrm{supp}(\boldsymbol{a}) := \{i \in [d] \mid \boldsymbol{a}_i \neq 0\}$ as the support set. We denote $\mathcal{V}_s^K := \{\boldsymbol{y} \in \mathbb{R}^K \mid \|\boldsymbol{y}\|_0 \leq s\}$ as a set of $s$-sparse vectors, $\mathbb{B}_2(\boldsymbol{c}; \rho) := \{\boldsymbol{y} \in \mathbb{R}^K \mid \|\boldsymbol{y} - \boldsymbol{c}\|_2 \leq \rho\}$ as a closed $\ell_2$-ball with center $\boldsymbol{c}$ and radius $\rho$, $\mathcal{N}(\mu, \sigma^2)$ as a Gaussian distribution with mean $\mu$ and covariance $\sigma^2$.

For two sequences of non-negative reals $\{f_n\}_{n \geq 1}$ and $\{g_n\}_{n \geq 1}$, we use $f_n \lesssim g_n$ to indicate that there is a universal constant $C > 0$ such that $f_n \leq C g_n$ for all $n \geq 1$. We use standard order notation $f_n = O(g_n)$ to indicate that $f_n \lesssim g_n$ and $f_n = \widetilde{O}_\tau(g_n)$ to indicate that $f_n \lesssim g_n \ln^c(1/\tau)$ for some universal constants $\tau$ and $c$. Throughout, we use $\epsilon, \delta, \tau, c, c_1, c_2, \ldots$ and $C, C_1, C_2, \ldots$ to denote universal positive constants, and their values may change from line to line without specific comments.

## 2 Literature Review

Multi-output regression (MOR) and its variants have been studied extensively over the past decades. In this section, we focus on existing works related to our computational and statistical results.

**Computational part.** Existing computational methods for solving MOR can be, in general, classified into two categories [8], known as problem transformation methods and algorithm adaptation methods. Problem transformation methods (e.g., Binary Relevance (BR), multi-target regressor stacking (MTRS) method [37], regression chains method [37]) aim to transform MOR into multiple single-output regression problems. Thus, any state-of-the-art single-output regression algorithm can be applied, such as ridge regression [15], regression trees [9], and etc. However, these transformation methods ignore the underlying structures/relations between outputs, which leads to higher computational complexities. In contrast, algorithm adaptation methods focus more on the underlying structures/relations between outputs. For instance, [36] investigates input component selection and shrinkage in multioutput linear regression; [1] later couples linear regressions and quantile mapping and thus captures joint relationships among variables. However, the output dimension considered in these works is relatively small compared with modern applications, and their assumptions concerning low-dimensional structure of outputs are hard to verify. *To overcome these shortages, we consider high-dimensional-output regression with only an additional sparsity requirement on outputs.*

**Statistical part.** There are numerous works concerning statistical properties of traditional or multi-output regressions. [18] gives sharp results on "out-of-sample" (random design) prediction error for the ordinary least squares estimator of traditional linear regression. [45] proposes an empirical risk minimization framework for large-scale multi-label learning with missing outputs and provides excess risk generalization error bounds with additional bounded constraints. [28] investigates the generalization performance of structured prediction learning and provides generalization error bounds on three different scenarios, i.e., Lipschitz continuity, smoothness, and space capacity condition. [27] designs an efficient feature selection procedure for multiclass sparse linear classifiers (a special case for SHORE with sparsity-level $s = 1$), and proves that the proposed classifiers guarantee the minimax generalization error bounds in theory. A recent paper [42] studies transfer learning via multi-task representation learning, a special case in MOR, which proves statistically optimistic rates on the excess risk with regularity assumptions on the loss function and task diversity. *In contrast with these works, our contributions concentrate on how generalization error bounds change before and after the compression under relatively weak conditions on the loss function and underlying distributions.*

**Specific results in MLC.** MLC is an important and special case for MOR with $\{0, 1\}$-valued output per dimension, i.e., $\boldsymbol{y} \in \{0, 1\}^K$, and thus, in this paragraph, we use labels to replace

outputs. Here, we focus on dimensionality reduction techniques on outputs, in particular, the compressed sensing and low-rank conditions on the output matrix $\boldsymbol{Y}$. The idea of compressed sensing rises from signal processing, which maps the original high-dimensional output space into a smaller one while ensuring the restricted isometry property (RIP). To the best of our knowledge, the compressed sensing technique is first used in [17] to handle a sparse expected output $\mathbb{E}[\boldsymbol{y}|\boldsymbol{x}]$. Later, [39, 12] propose Principle Label Space Transformation (PLST) and conditional PLST through singular value decomposition and canonical component analysis respectively. More recently, many new compression approaches have been proposed, such as robust bloom filter [13], log time log space extreme classification [23], merged averaged classifiers via hashing [32], etc. Additionally, computational efficiency and statistical generalization bounds can be further improved when the output matrix $\boldsymbol{Y}$ ensures a low-rank condition. Under such a condition, [45] provides a general empirical risk minimization framework for solving MLC with missing labels. *Compared with the above works, this paper studies MOR under a sparse & high-dimensional-output setting without additional correlation assumptions or low-rank assumptions for output space, and then provides a complete story through a computational and statistical lens.*

## 3 Main Results

### 3.1 Two-Stage Framework

This subsection presents a general framework for solving SHORE and then the computational complexity for the proposed framework *with/without compression*. Given a set of training samples $\{(\boldsymbol{x}^i, \boldsymbol{y}^i)\}_{i=1}^n$ as described in Section 1, the framework can be separated into two stages: (compressed) training stage & (compressed) prediction stage.

**Training stage.** In the first training stage, the framework finds a *compressed regressor* by solving a linear regression problem with compressed outputs. In particular, the framework compresses the original large output space ($K$-dim) to a smaller "latent" output space ($m$-dim) by left-multiplying a so-called "compressed" matrix $\boldsymbol{\Phi} \in \mathbb{R}^{m \times K}$ to outputs. Thus, the *compressed version* of training stage in SHORE can be represented as follows,

$$\widehat{\boldsymbol{W}} := \underset{\boldsymbol{W} \in \mathbb{R}^{m \times d}}{\arg\min} \ \widehat{\mathcal{L}}_n^{\boldsymbol{\Phi}}(\boldsymbol{W}) := \frac{1}{n}\|\boldsymbol{\Phi}\boldsymbol{Y} - \boldsymbol{W}\boldsymbol{X}\|_F^2. \tag{2}$$

We would like to point out that the idea of compressing the output space into some smaller intrinsic dimension has been used in many existing works, e.g., [17, 39, 12] mentioned in Section 2.

**Prediction stage.** In the second prediction stage, given any input $\boldsymbol{x} \in \mathbb{R}^d$, the framework predicts a sparse output $\widehat{\boldsymbol{y}}$ by solving the following prediction problem based on the learned regressor $\widehat{\boldsymbol{W}}$ in the training stage,

$$\widehat{\boldsymbol{y}}(\widehat{\boldsymbol{W}}) := \underset{\boldsymbol{y}}{\arg\min} \ \|\boldsymbol{\Phi}\boldsymbol{y} - \widehat{\boldsymbol{W}}\boldsymbol{x}\|_2^2 \ \text{ s.t. } \ \boldsymbol{y} \in \mathcal{V}_s^K \cap \mathcal{F}, \tag{3}$$

where $\mathcal{V}_s^K$ is the set of $s$-sparse vectors in $\mathbb{R}^K$, and $\mathcal{F}$ is some feasible set to describe additional requirements of $\boldsymbol{y}$. For example, by letting $\mathcal{F}$ be $\mathbb{R}^K, \mathbb{R}_+^K, \{0, 1\}^K$, the intersection $\mathcal{V}_s^K \cap \mathcal{F}$ denotes the set of $s$-sparse output, non-negative $s$-sparse output, $\{0, 1\}$-valued $s$-sparse output, respectively. We use $\widehat{\boldsymbol{y}}(\widehat{\boldsymbol{W}})$ (shorthanded in $\widehat{\boldsymbol{y}}$) to specify that the predicted output is based on the regressor $\widehat{\boldsymbol{W}}$. To solve the proposed prediction problem (3), we utilize the following projected gradient descent method (Algorithm 1), which could be viewed as a variant/generalization of existing iterative thresholding methods [6, 21] for nonconvex constrained minimization. In particular, step 4 incorporates additional constraints from $\mathcal{F}$ other than sparsity into consideration, which leads to non-trivial modifications in designing efficient projection oracles and convergence analysis. Later, we show that the proposed Algorithm 1 ensures a near-optimal convergence (Theorem 2 and Theorem 4) while greatly reduces the computational complexity (Remark 2) of the prediction stage for solving compressed SHORE.

Before diving into theoretical analysis, we first highlight the differences between the proposed prediction stage (3), general sparsity-constrained optimization (SCO), and sparse regression in the following remark.

**Remark 1.** *Proposed prediction stage v.s. General SCO: To be clear, the SCO here denotes the following minimization problem $\min_{\|\boldsymbol{\alpha}\|_0 \leq k} \|\boldsymbol{A}\boldsymbol{\alpha} - \boldsymbol{\beta}\|_2^2$. Thus, the prediction stage is a special case*

*of general SCO problem. In particular, the predicted stage takes a random projection matrix $\boldsymbol{\Phi}$ with restricted isometry property (RIP) to be its $\boldsymbol{A}$ and uses $\widehat{\boldsymbol{W}}\boldsymbol{x}$ with $\widehat{\boldsymbol{W}}$ obtained from the compressed training-stage to be its $\boldsymbol{\beta}$. As a result (Theorem 2 and Theorem 4), the proposed Algorithm 1 for prediction stage ensures a globally linear convergence to a ball with center $\widehat{\boldsymbol{y}}$ (optimal solution of the prediction-stage) and radius $O(\|\boldsymbol{\Phi}\widehat{\boldsymbol{y}} - \widehat{\boldsymbol{W}}\boldsymbol{x}\|_2)$, which might not hold for general SCO problems.*

***Proposed prediction stage v.s. Sparse regression:*** *Although the proposed prediction stage and sparse high-dimensional regression share a similar optimization formulation $\min_{\|\boldsymbol{\beta}\|_0 \leq k} \ \|\boldsymbol{Y} - \boldsymbol{X}^\top \boldsymbol{\beta}\|_2^2$, the proposed prediction stage (3) is distinct from the sparse regression in the following parts:*

***(1) Underlying Model:*** *Most existing works about sparse high-dimensional regression assume that samples are i.i.d. generated from the linear relationship $\boldsymbol{Y} = \boldsymbol{X}^\top \boldsymbol{\beta}^* + \boldsymbol{\epsilon}$ with underlying sparse ground truth $\boldsymbol{\beta}^*$. In the proposed prediction stage, we do not assume additional underlying models on samples if there is no further specific assumption. The problem we studied in the predicted stage takes the random projection matrix $\boldsymbol{\Phi}$ with restricted isometry property (RIP) as its $\boldsymbol{X}^\top$ (whereas $\boldsymbol{X}^\top$ in sparse regression does not ensure RIP), and uses $\widehat{\boldsymbol{W}}\boldsymbol{x}$ with $\widehat{\boldsymbol{W}}$ obtained from the compressed training-stage as its $\boldsymbol{Y}$.*

***(2) Problem Task:*** *The sparse regression aims to recover the sparse ground truth $\boldsymbol{\beta}^*$ given a sample set $\{(\boldsymbol{x}^i, \boldsymbol{y}^i)\}_{i=1}^n$ with $n$ i.i.d. samples. In contrast, the task of the proposed prediction stage is to predict a sparse high-dimensional output $\widehat{\boldsymbol{y}}$ given a random projection matrix $\boldsymbol{\Phi}$ and a single input $\boldsymbol{x}$. As a quick summary, some typical and widely used iterative algorithms [38, 3, 4, 29] for sparse regression cannot be directly applied to the proposed prediction stage.*

Then, we provide the computational complexity *with and without the compression* for the proposed two-stage framework to complete this subsection.

**Remark 2.** ***Training stage:*** *Conditioned on $\boldsymbol{X}\boldsymbol{X}^\top$ is invertible, the compressed regressor $\widehat{\boldsymbol{W}}$ has a closed form solution $\widehat{\boldsymbol{W}} = \boldsymbol{\Phi}\boldsymbol{Y}\boldsymbol{X}^\top(\boldsymbol{X}\boldsymbol{X}^\top)^{-1}$ with overall computational complexity*

$$O(Kmn + mnd + nd^2 + d^3 + md^2) \approx O(Kmn).$$

*Compared with the computational complexity of finding $\widehat{\boldsymbol{Z}}$ from the uncompressed SHORE (1)*

$$O(Knd + nd^2 + d^3 + Kd^2) \approx O(K(n+d)d),$$

*solving $\widehat{\boldsymbol{W}}$ enjoys a smaller computational complexity on the training stage if $m \ll d$. In later analysis (see Section 3.2), $m = O(\delta^{-2} \cdot s \log(\frac{K}{\tau}))$ with some predetermined constants $\delta, \tau$ and sparsity-level $s \ll d$, thus in many applications with large output space, the condition $m \ll d$ holds.*

***Prediction stage:*** *The computational complexity of each step-3 in Algorithm 1 is*

$$O(Km + K + Km + K) \approx O(Km).$$

*The projection in step-4 is polynomially solvable with computational complexity $O(K \min\{s, \log K\})$ (see proof in Appendix A.5.1). Thus, the overall computational complexity of Algorithm 1 is*

$$O(K(m + \min\{s, \log K\})T).$$

*Compared with the complexity $O(K(d + \min\{s, \log K\}))$ of predicting $\widehat{\boldsymbol{y}}$ from the uncompressed SHORE (1), the compressed version enjoys a smaller complexity on the prediction stage if*

$$(m + \min\{s, \log K\})T \ll d + \min\{s, \log K\}. \tag{4}$$

*In later analysis (see Theorem 2), since $m = O(\delta^{-2} \cdot s \log(\frac{K}{\tau}))$ with predetermined constants $\delta, \tau$, sparsity-level $s \ll d$, and $T = O(\log[\frac{\|\widehat{\boldsymbol{y}} - \boldsymbol{v}^{(0)}\|_2}{\|\boldsymbol{\Phi}\widehat{\boldsymbol{y}} - \widehat{\boldsymbol{W}}\boldsymbol{x}\|_2}])$ from inequality (5) , we have condition 4 holds.*

***Whole computational complexity:*** *Based on the analysis of computational complexity above, we conclude that when the parameters $(K, d, m, T)$ satisfies*

$$K > K^{1/3} > d \gg O(\delta^{-2} \log(K/\tau) \cdot T) = mT,$$

*the compressed SHORE enjoys a better computational complexity with respect to the original one (1).*

---

**Algorithm 1** Projected Gradient Descent (for Second Stage)

---

**Input:** Regressor $\widehat{W}$, input sample $x$, stepsize $\eta$, total iterations $T$

1: **Initialize** point $v^{(0)} \in \mathcal{V}_s^K \cap \mathcal{F}$.
2: **for** $t = 0, 1, \ldots, T - 1$: **do**
3:     Update $\widetilde{v}^{(t+1)} = v^{(t)} - \eta \cdot \Phi^\top (\Phi v^{(t)} - \widehat{W} x)$.
4:     Project $v^{(t+1)} = \Pi(\widetilde{v}^{(t+1)}) := \arg\min_{v \in \mathcal{V}_s^K \cap \mathcal{F}} \|v - \widetilde{v}^{(t+1)}\|_2^2$.
5: **end for**

**Output:** $v^{(T)}$.

---

### 3.2 Worst-Case Analysis for Arbitrary Samples

We begin this subsection by introducing the generalization method of the compressed matrix $\Phi$.

**Assumption 1.** *Given an $m$-by-$K$ compressed matrix $\Phi$, all components $\Phi_{i,j}$ for $1 \leq i \leq m$ and $1 \leq j \leq K$, are i.i.d. generated from a Gaussian distribution $\mathcal{N}(0, 1/m)$.*

Before presenting the main theoretical results, let us first introduce the definition of restricted isometry property (RIP, [10]), which is ensured by the generalization method (Assumption 1).

**Definition 1.** *$(\mathcal{V}, \delta)$-**RIP**: A $m$-by-$K$ matrix $\Phi$ is said to be $(\mathcal{V}, \delta)$-RIP over a given set of vectors $\mathcal{V} \subseteq \mathbb{R}^K$, if, for every $v \in \mathcal{V}$,*

$$(1 - \delta)\|v\|_2^2 \leq \|\Phi v\|_2^2 \leq (1 + \delta)\|v\|_2^2.$$

*In the rest of the paper, we use $(s, \delta)$-RIP to denote $(\mathcal{V}_s^K, \delta)$-RIP. Recall $\mathcal{V}_s^K = \{v \in \mathbb{R}^K \mid \|v\|_0 \leq s\}$ is the set of $s$-sparse vectors.*

**Remark 3.** *From Johnson-Lindenstrauss Lemma [43], for any $\delta \in (0, 1)$, any $\tau \in (0, 1)$, and any finite vector set $|\mathcal{V}| < \infty$, if the number of rows $m \geq O\left(\delta^{-2} \cdot \log(\frac{|\mathcal{V}|}{\tau})\right)$, then the compressed matrix $\Phi$ generated by Assumption 1 satisfies $(\mathcal{V}, \delta)$-RIP with probability at least $1 - \tau$.*

Now, we are poised to present the first result on training loss defined in (2).

**Theorem 1.** *For any $\delta \in (0, 1)$ and $\tau \in (0, 1)$, suppose compressed matrix $\Phi$ follows Assumption 1 with $m \geq O(\frac{1}{\delta^2} \cdot \log(\frac{K}{\tau}))$. We have the following inequality for training loss*

$$\|\Phi Y - \widehat{W} X\|_F^2 \leq (1 + \delta) \cdot \|Y - \widehat{Z} X\|_F^2,$$

*holds with probability at least $1 - \tau$, where $\widehat{Z}, \widehat{W}$ are optimal solutions for the uncompressed (1) and compressed SHORE (2), respectively.*

The proof of Theorem 1 is presented in Appendix A.1. In short, Theorem 1 shows that the optimal training loss for the compressed version is upper bounded within a $(1 + \delta)$ multiplicative ratio with respect to the optimal training loss for the uncompressed version. Intuitively, Theorem 1 implies that SHORE remains similar performances for both compressed and compressed versions, while the compressed version saves roughly $O(Kn(d - m) + Kd^2)$ computational complexity in the training stage from Remark 2. Moreover, the lower bound condition on $m \geq O(\frac{1}{\delta^2} \cdot \log(\frac{K}{\tau}))$ ensures that the generated compressed matrix $\Phi$ is $(1, \delta)$-RIP with probability at least $1 - \tau$. For people of independent interest, Theorem 1 only needs $(1, \delta)$-RIP (independent with the sparsity level) due to the *unitary invariant* property of $\Phi$ from Assumption 1 (details in Appendix A.1). Additionally, due to the inverse proportionality between $m$ and $\delta^2$, for fixed $K$ and $\tau$, the result can be written as

$$\|\Phi Y - \widehat{W} X\|_F^2 \leq \left(1 + O(1/\sqrt{m})\right) \cdot \|Y - \widehat{Z} X\|_F^2,$$

which is verified in our experiments 4.

We then present the convergence result of the proposed Algorithm 1 for solving prediction problem (3).

**Theorem 2.** *For any $\delta \in (0, 1)$ and $\tau \in (0, 1)$, suppose the compressed matrix $\Phi$ follows Assumption 1 with $m \geq O(\frac{s}{\delta^2} \log(\frac{K}{\tau}))$. With a fixed stepsize $\eta \in (\frac{1}{2 - 2\delta}, 1)$, the following inequality*

$$\|\widehat{y} - v^{(t)}\|_2 \leq c_1^t \cdot \|\widehat{y} - v^{(0)}\|_2 + \frac{c_2}{1 - c_1} \cdot \|\Phi \widehat{y} - \widehat{W} x\|_2$$

*holds for all $t \in [T]$ simultaneously with probability at least $1 - \tau$, where $c_1 := 2 - 2\eta + 2\eta\delta < 1$ is some positive constant strictly smaller than 1, and $c_2 := 2\eta\sqrt{1 + \delta}$ is some constant.*

The proof of Theorem 2 is given in Appendix A.2. Here, the lower bound condition on the number of rows $m$ ensures that the generated compressed matrix $\mathbf{\Phi}$ is $(3s, \delta)$-RIP with probability at least $1 - \tau$ by considering a $\delta/2$-net cover of set $\mathcal{V} = \mathcal{V}_{3s}^K \cap \mathbb{B}_2(\mathbf{0}; 1)$ from Johnson-Lindenstrauss Lemma [43]. Moreover, since the number of rows $m$ required in Theorem 2 is greater than the one required in Theorem 1, term $\|\mathbf{\Phi}\widehat{\boldsymbol{y}} - \widehat{\boldsymbol{W}}\boldsymbol{x}\|_2$ can be further upper bounded using the uncompressed version $(1 + \delta)\|\widehat{\boldsymbol{y}} - \widehat{\boldsymbol{Z}}\boldsymbol{x}\|_2$ with probability at least $1 - \tau$. Then we obtain a direct corollary of Theorem 2: suppose $\|\widehat{\boldsymbol{y}} - \boldsymbol{v}^{(0)}\|_2 > \|\mathbf{\Phi}\widehat{\boldsymbol{y}} - \widehat{\boldsymbol{W}}\boldsymbol{x}\|_2$, if

$$t \geq t_* := O\left(\log\left(\|\widehat{\boldsymbol{y}} - \boldsymbol{v}^{(0)}\|_2 / \|\mathbf{\Phi}\widehat{\boldsymbol{y}} - \widehat{\boldsymbol{W}}\boldsymbol{x}\|_2\right)\Big/ \log\left(1/c_1\right)\right), \tag{5}$$

the proposed Algorithm 1 guarantees a globally linear convergence to a ball $\mathbb{B}(\widehat{\boldsymbol{y}}; O(\|\mathbf{\Phi}\widehat{\boldsymbol{y}} - \widehat{\boldsymbol{W}}\boldsymbol{x}\|_2))$.

In contrast with OMP used in [17] for multi-label predictions, Theorem 2 holds for arbitrary sample set without the so-called bounded coherence guarantee on $\mathbf{\Phi}$. Moreover, as reported in Section 4, the proposed prediction method (Algorithm 1) has better computational efficiency than OMP.

### 3.3 Generalization Error Bounds for IID Samples

This subsection studies a specific scenario when every sample $(\boldsymbol{x}^i, \boldsymbol{y}^i)$ is i.i.d. drawn from some underlying subGaussian distribution $\mathcal{D}$ over sample space $\mathbb{R}^d \times \mathcal{V}_s^K$. Specifically, we use

$$\mathbb{E}_{\mathcal{D}}\left[\begin{pmatrix}\boldsymbol{x} \\ \boldsymbol{y}\end{pmatrix}\right] = \begin{pmatrix}\boldsymbol{\mu_x} \\ \boldsymbol{\mu_y}\end{pmatrix} =: \boldsymbol{\mu} \quad \text{and} \quad \text{Var}_{\mathcal{D}}\left[\begin{pmatrix}\boldsymbol{x} \\ \boldsymbol{y}\end{pmatrix}\right] = \begin{pmatrix}\boldsymbol{\Sigma_{xx}} & \boldsymbol{\Sigma_{xy}} \\ \boldsymbol{\Sigma_{yx}} & \boldsymbol{\Sigma_{yy}}\end{pmatrix} =: \boldsymbol{\Sigma} \succeq \mathbf{0}_{(d+K)\times(d+K)}$$

to denote its mean and variance, respectively. Let $\boldsymbol{\xi_x} := \boldsymbol{x} - \boldsymbol{\mu_x}, \boldsymbol{\xi_y} := \boldsymbol{y} - \boldsymbol{\mu_y}, \boldsymbol{\xi} := (\boldsymbol{\xi_x}^\top, \boldsymbol{\xi_y}^\top)^\top$ be centered subGaussian random variables of $\boldsymbol{x}, \boldsymbol{y}, (\boldsymbol{x}^\top, \boldsymbol{y}^\top)^\top$, respectively. Let $\boldsymbol{a}, \boldsymbol{b} \in \{\boldsymbol{x}, \boldsymbol{y}\}$, we use $\boldsymbol{M_{ab}} := \mathbb{E}_{\mathcal{D}}[\boldsymbol{a}\boldsymbol{b}^\top] = \boldsymbol{\Sigma_{ab}} + \boldsymbol{\mu_a}\boldsymbol{\mu_b}^\top, \widehat{\boldsymbol{M}}_{\boldsymbol{ab}} := \frac{1}{n}\sum_{i=1}^n \boldsymbol{a}^i(\boldsymbol{b}^i)^\top$ to denote the population second (cross-)moments and empirical second (cross-)moments, respectively. Then, the population training loss is defined as

$$\min_{\boldsymbol{Z} \in \mathbb{R}^{K \times d}} \mathcal{L}(\boldsymbol{Z}) := \mathbb{E}_{(\boldsymbol{x}, \boldsymbol{y}) \sim \mathcal{D}}\left[\|\boldsymbol{y} - \boldsymbol{Z}\boldsymbol{x}\|_2^2\right]$$

with its optimal solution $\boldsymbol{Z}_* := \boldsymbol{M_{yx}}\boldsymbol{M_{xx}}^{-1}$. Similarly, given a $\mathbf{\Phi}$, the compressed training loss is given by

$$\mathcal{L}^{\mathbf{\Phi}}(\boldsymbol{W}) := \mathbb{E}_{(\boldsymbol{x}, \boldsymbol{y}) \sim \mathcal{D}}\left[\|\mathbf{\Phi}\boldsymbol{y} - \boldsymbol{W}\boldsymbol{x}\|_2^2\right]$$

with optimal solution $\boldsymbol{W}_* := \mathbf{\Phi}\boldsymbol{M_{yx}}\boldsymbol{M_{xx}}^{-1}$. We then define the following assumption:

**Assumption 2.** *Let $\mathcal{D}$ be $\sigma^2$-subGaussian for some positive constant $\sigma^2 > 0$, i.e., the inequality $\mathbb{E}_{\mathcal{D}}[\exp(\lambda \boldsymbol{u}^\top \boldsymbol{\xi})] \leq \exp\left(\lambda^2 \sigma^2 / 2\right)$ holds for any $\lambda > 0$ and unitary vector $\boldsymbol{u} \in \mathbb{R}^{d+K}$. Moreover, the covariance matrix $\boldsymbol{\Sigma_{xx}}$ is positive definite (i.e., its minimum eigenvalue $\lambda_{\min}(\boldsymbol{\Sigma_{xx}}) > 0$).*

**Remark 4.** *Assumption 2 ensures the light tail property of distribution $\mathcal{D}$. Note that in some real applications, e.g., factuality check [31], algorithmic trading [16], one can normalize input and output vector to ensure bounded $\ell_2$-norm. Under such a situation, Assumption 2 is naturally satisfied.*

Our first result in this subsection gives the generalization error bounds.

**Theorem 3.** *For any $\delta \in (0, 1)$ and $\tau \in (0, \frac{1}{3})$, suppose compressed matrix $\mathbf{\Phi}$ follows Assumption 1 with $m \geq O(\frac{s}{\delta^2}\log(\frac{K}{\tau}))$, and Assumption 2 holds, for any constant $\epsilon > 0$, the following results hold:*

*(Matrix Error). The inequality for matrix error $\left\|\boldsymbol{M_{xx}}^{1/2}\widehat{\boldsymbol{M}}_{\boldsymbol{xx}}^{-1}\boldsymbol{M_{xx}}^{1/2}\right\|_{op} \leq 4$ holds with probability at least $1 - 2\tau$ as the number of samples $n \geq n_1$ with*

$$n_1 := \max\left\{\frac{64C^2\sigma^4}{9\lambda_{\min}^2(\boldsymbol{M_{xx}})}\left(d + \log(2/\tau)\right), \frac{32^2\|\boldsymbol{\mu_x}\|_2^2\sigma^2}{\lambda_{\min}^2(\boldsymbol{M_{xx}})}\left(2\sqrt{d} + \sqrt{\log(1/\tau)}\right)^2\right\},$$

*where $C$ is some fixed positive constant used in matrix concentration inequality of operator norm.*

*(Uncompressed). The generalization error bound for uncompressed SHORE satisfies $\mathcal{L}(\widehat{\boldsymbol{Z}}) \leq \mathcal{L}(\boldsymbol{Z}_*) + 4\epsilon$ with probability at least $1 - 3\tau$, as the number of samples $n \geq \max\{n_1, n_2\}$ with*

$$n_2 := \max\left\{4(\|\boldsymbol{Z}_*\|_F^2 + K) \cdot \frac{d + 2\sqrt{d\log(K/\tau)} + 2\log(K/\tau)}{\epsilon}, \; 4\|\boldsymbol{\mu_y} - \boldsymbol{Z}_*\boldsymbol{\mu_x}\|_2^2 \cdot \frac{d}{\epsilon}\right\}.$$

*(Compressed). The generalization error bound for the compressed SHORE satisfies $\mathcal{L}^{\Phi}(\widehat{W}) \leq \mathcal{L}^{\Phi}(W_*) + 4\epsilon$ with probability at least $1 - 3\tau$, as the number of sample $n \geq \max\{n_1, \widetilde{n}_2\}$ with*

$$\widetilde{n}_2 := \max\left\{ 4(\|W_*\|_F^2 + \|\Phi\|_F^2) \cdot \frac{d + 2\sqrt{d\log(m/\tau)} + 2\log(m/\tau)}{\epsilon}, \ 4\|\Phi\mu_y - W_*\mu_x\|_2^2 \cdot \frac{d}{\epsilon} \right\}.$$

The proof of Theorem 3 is presented in Appendix A.3. The proof sketch mainly contains three steps: In *Step-1*, we represent the difference $\mathcal{L}(\widehat{Z}) - \mathcal{L}(Z_*)$ or $\mathcal{L}^{\Phi}(\widehat{W}) - \mathcal{L}^{\Phi}(W_*)$ as a product between matrix error (in Theorem 3) and rescaled approximation error (see Appendix A.3 for definition), i.e.,

$$\mathcal{L}(\widehat{Z}) - \mathcal{L}(Z_*) \ \text{ or } \ \mathcal{L}^{\Phi}(\widehat{W}) - \mathcal{L}^{\Phi}(W_*) \leq (\text{matrix error}) \times (\text{rescaled approximation error});$$

*Step-2* controls the upper bounds for matrix error and rescaled approximation error separately, using concentration for subGuassian variables; for *Step-3*, we combine the upper bounds obtained in Step-2 and complete the proof. Based on the result of Theorem 3, ignoring the logarithm term for $\tau$, the proposed generalization error bounds can be bounded by

$$\mathcal{L}(\widehat{Z}) \leq \mathcal{L}(Z_*) + \widetilde{O}_\tau\left( \max\{\|Z_*\|_F^2, \ \|\mu_y - Z_*\mu_x\|_2^2, \ K\} \cdot \frac{d}{n} \right),$$

$$\mathcal{L}^{\Phi}(\widehat{W}) \leq \mathcal{L}^{\Phi}(W_*) + \widetilde{O}_\tau\left( \max\{\|W_*\|_F^2, \ \|\Phi\mu_y - W_*\mu_x\|_2^2, \ \|\Phi\|_F^2\} \cdot \frac{d}{n} \right).$$

**Remark 5.** *To make a direct comparison between the generalization error bounds of the uncompressed and the compressed version, we further control the norms $\|W_*\|_F^2, \|\Phi\mu_y - W_*\mu_x\|_2^2, \|\Phi\|_F^2$ based on additional conditions on the compressed matrix $\Phi$. Recall the generalization method of the compressed matrix $\Phi$ as mentioned in Assumption 1, we have the following event*

$$\mathcal{E}_1 := \left\{ \Phi \in \mathbb{R}^{m \times K} \ \middle| \ \begin{array}{l} \|W_*\|_F^2 = \|\Phi Z_*\|_F^2 \leq (1+\delta)\|Z_*\|_F^2 \\ \|\Phi\mu_y - W_*\mu_x\|_F^2 = \|\Phi(\mu_y - Z_*\mu_x)\|_F^2 \leq (1+\delta)\|\mu_y - Z_*\mu_x\|_F^2 \end{array} \right\}$$

*holds with probability at least $1 - \tau$ due to the RIP property for a fixed matrix. Moreover, since every component $\Phi_{i,j}$ is i.i.d. drawn from a Gaussian distribution $\mathcal{N}(0, 1/m)$, using the concentration tail bound for chi-squared variables (See Lemma 1 in [26]), we have the following event*

$$\mathcal{E}_2 := \left\{ \Phi \in \mathbb{R}^{m \times K} \ \middle| \ \|\Phi\|_F^2 \leq K + 2\sqrt{\frac{K\log(1/\tau)}{m}} + \frac{2\log(1/\tau)}{m} \right\}$$

*holds with probability at least $1 - \tau$. Conditioned on these two events $\mathcal{E}_1$ and $\mathcal{E}_2$, the generalization error bound of the compressed version achieves the same order (ignoring the logarithm term of $\tau$) as the generalization error bound of the uncompressed version. That is to say,*

$$\mathcal{L}^{\Phi}(\widehat{W}) \leq (1+\delta) \cdot \mathcal{L}(Z_*) + \widetilde{O}_\tau\left( \max\{\|Z_*\|_F^2, \ \|\mu_y - Z_*\mu_x\|_2^2, \ K\} \cdot \frac{d}{n} \right)$$

*holds with probability at least $1 - 5\tau$.*

Comparing with existing results on generalization error bounds mentioned in Section 2, we would like to emphasize that Theorem 4 guarantees that the generalization error bounds maintain the order before and after compression. This result establishes on i.i.d. subGaussian samples for the SHORE model without additional regularity conditions on loss function and feasible set as required in [45]. Additionally, we obtained a $O(Kd/n)$ generalization error bound for squared Frobenius norm loss function $\mathcal{L}$ or $\mathcal{L}^{\Phi}$, which is smaller than $O(K^2 d/n)$ as presented in [Theorem 4, [45]].

We then give results on prediction error bounds.

**Theorem 4.** *For any $\delta \in (0,1)$ and any $\tau \in (0, 1/3)$, suppose the compressed matrix $\Phi$ follows Assumption 1 with $m \geq O(\frac{s}{\delta^2}\log(\frac{K}{\tau}))$, and Assumption 2 holds. Given any learned regressor $\widehat{W}$ from training problem (2), let $(x, y)$ be a new sample drawn from the underlying distribution $\mathcal{D}$, we have the following inequality holds with probability at least $1 - \tau$:*

$$\mathbb{E}_{\mathcal{D}}[\|\widehat{y} - y\|_2^2] \leq \frac{4}{1-\delta} \cdot \mathbb{E}_{\mathcal{D}}[\|\Phi y - \widehat{W}x\|_2^2],$$

*where $\widehat{y}$ is the optimal solution from prediction problem (3) with input vector $x$.*

The proof of Theorem 4 is presented in Appendix A.4. Theorem 4 gives an upper bound of $\ell_2$-norm distance between $\widehat{\boldsymbol{y}}$ and $\boldsymbol{y}$. Since $\|\boldsymbol{v}^{(T)} - \boldsymbol{y}\|_2 \leq \|\boldsymbol{v}^{(T)} - \widehat{\boldsymbol{y}}\|_2 + \|\widehat{\boldsymbol{y}} - \boldsymbol{y}\|_2$, combined with Theorem 2, we have $\mathbb{E}_{\mathcal{D}}[\|\boldsymbol{v}^{(T)} - \boldsymbol{y}\|_2^2] \leq O(\mathbb{E}_{\mathcal{D}}[\|\boldsymbol{\Phi}\boldsymbol{y} - \widehat{\boldsymbol{W}}\boldsymbol{x}\|_2])$ when $T \geq t_*$ defined in (5) (see Appendix A.5.5), where the final inequality holds due to the optimality of $\widehat{\boldsymbol{y}}$. Hence, we achieve an upper bound of $\ell_2$-norm distance between $\boldsymbol{v}^{(T)}$ and $\boldsymbol{y}$ as presented in Theorem 4, see Remark 6.

**Remark 6.** . *For any $\delta \in (0,1)$ and $\tau \in (0, 1/3)$, suppose compressed matrix $\boldsymbol{\Phi}$ follows Assumption 1 with $m \geq O(\frac{s}{\delta^2} \log(\frac{K}{\tau}))$, and Assumption 2 holds, for any constant $\epsilon > 0$, the following inequality holds with probability at least $1 - 3\tau$:*

$$\mathbb{E}_{\mathcal{D}}[\|\boldsymbol{v}^{(T)} - \boldsymbol{y}\|_2^2] \leq O(\mathbb{E}_{\mathcal{D}}[\|\boldsymbol{\Phi}\boldsymbol{y} - \widehat{\boldsymbol{W}}\boldsymbol{x}\|_2]) \leq O(\mathcal{L}^{\boldsymbol{\Phi}}(\boldsymbol{W}_*) + 4\epsilon).$$

## 4 Numerical Experiments

In this section, we conduct numerical experiments on two types of instances (i.e., synthetic data sets and real data sets) to validate the theoretical results and illustrate both efficiency and accuracy of the proposed prediction method compared with typical existing prediction baselines, i.e., Orthogonal Matching Pursuit (OMP, [46]), Fast Iterative Shrinkage-Thresholding Algorithm (FISTA, [2]) and Elastic Net (EN, [47]). Due to the space limit, we put the implemented prediction method (Algorithm 2) in Appendix A.6.1, aforementioned existing prediction baselines in Appendix A.6.2, experiment setting details and results for real data in Appendix A.7.

**Performance measures.** Given a sample $(\boldsymbol{x}, \boldsymbol{y})$ with input $\boldsymbol{x}$ and corresponding true output $\boldsymbol{y}$, we use $\boldsymbol{v}$ to denote the predicted output obtained from any prediction method, and measure the numerical performances based on the following three metrics:

1. For a ground truth $\boldsymbol{y}$ with sparsity-level $s$, the metric *precision over selected supports*, i.e., $\texttt{Precision@s} := \frac{1}{s}|\text{supp}(\boldsymbol{v}) \cap \text{supp}(\boldsymbol{y})|$ measures the percentage of correctly identified supports in the predicted output;

2. The metric *output difference*, i.e., $\texttt{Output} - \texttt{diff} := \|\boldsymbol{v} - \boldsymbol{y}\|_2^2$ measures the $\ell_2$-norm distance between the predicted output and the ground-truth;

3. For any given MOR $\widehat{\boldsymbol{W}}$ and compressed matrix $\boldsymbol{\Phi}$, the metric *prediction loss*, i.e., $\texttt{Prediction} - \texttt{Loss} := \|\boldsymbol{\Phi}\boldsymbol{v} - \widehat{\boldsymbol{W}}\boldsymbol{x}\|_2^2$ computes the prediction loss with respect to $\widehat{\boldsymbol{W}}\boldsymbol{x}$.

**Synthetic data generation procedure.** The synthetic data set is generated as follows: Every input $\boldsymbol{x}^i$ for $i \in [n]$ is i.i.d. drawn from a Gaussian distribution $\mathcal{N}(\boldsymbol{\mu}_{\boldsymbol{x}}, \boldsymbol{\Sigma}_{\boldsymbol{xx}})$, where its mean vector $\boldsymbol{\mu}_{\boldsymbol{x}}$ and covariance matrix $\boldsymbol{\Sigma}_{\boldsymbol{xx}}$ are selected based on the procedures given in Appendix A.6.3. For any given sparsity-level $s$, underlying true regressor $\boldsymbol{Z}_* \in \mathbb{R}^{K \times d}$, and Signal-to-Noise Ratio (SNR), the ground-truth $\boldsymbol{y}^i$ (corresponding with its given input $\boldsymbol{x}^i$) is generated by $\boldsymbol{y}^i = \Pi_{\mathcal{V}_s^K \cap \mathcal{F}} (\boldsymbol{Z}_* \boldsymbol{x}^i + \boldsymbol{\epsilon}^i)$, where $\boldsymbol{\epsilon}^i \in \mathbb{R}^K$ is a i.i.d. random noise drawn from the Gaussian distribution $\mathcal{N}(\boldsymbol{0}_K, \text{SNR}^{-2}\|\boldsymbol{Z}_*\boldsymbol{x}^i\|_\infty \cdot \boldsymbol{I}_K)$.

**Parameter setting.** For synthetic data, we set input dimension $d = 10^4$, output dimension $K = 2 \times 10^4$, and sparsity-level $s = 3$. We generate in total $n = 3 \times 10^4$, i.i.d. samples as described above, i.e., $\mathcal{S}^{\texttt{syn}} := \{(\boldsymbol{x}^i, \boldsymbol{y}^i)\}_{i=1}^{3 \times 10^4}$ with $\text{SNR}^{-1} \in \{1, 0.32, 0.032\}$ to ensure the signal-to-noise decibels (dB, [14]) takes values on $\text{dB} := 10\log(\text{SNR}^2) \in \{0, 10, 30\}$. We select the number of rows for compressed matrix $\boldsymbol{\Phi}$ by $m \in \{100, 300, 500, 700, 1000, 2000\}$. For computing the empirical regressor $\widehat{\boldsymbol{W}} \in \mathbb{R}^{m \times d}$, we first split the whole sample set $\mathcal{S}^{\texttt{syn}}$ into two non-overlap subsets, i.e., a training set $\mathcal{S}^{\texttt{tra}}$ with 80% and a testing set $\mathcal{S}^{\texttt{test}}$ with rest 20%. The regressor $\widehat{\boldsymbol{W}}$ is therefore obtained by solving compressed SHORE (2) based on the training set $\mathcal{S}^{\texttt{tra}}$ with a randomly generated compressed matrix $\boldsymbol{\Phi}$. For evaluating the proposed prediction method, Algorithm 2, we pick a fixed stepsize $\eta = 0.9$, $\mathcal{F} = \mathbb{R}_+^K$, and set the maximum iteration number as $T = 60$, and run prediction methods over the set $\mathcal{S}^{\texttt{test}}$.

**Hardware & Software.** All experiments are conducted in Dell workstation Precision 7920 with a 3GHz 48Cores Intel Xeon CPU and 128GB 2934MHz DDR4 Memory. The proposed method and other methods are solved using PyTorch version 2.3.0 and scikit-learn version 1.4.2 in Python 3.12.3.

**Numerical Results & Discussions.** The results are demonstrated in Figure 1, which does not include the results from the Elastic Net and OMP due to relatively much longer running time.

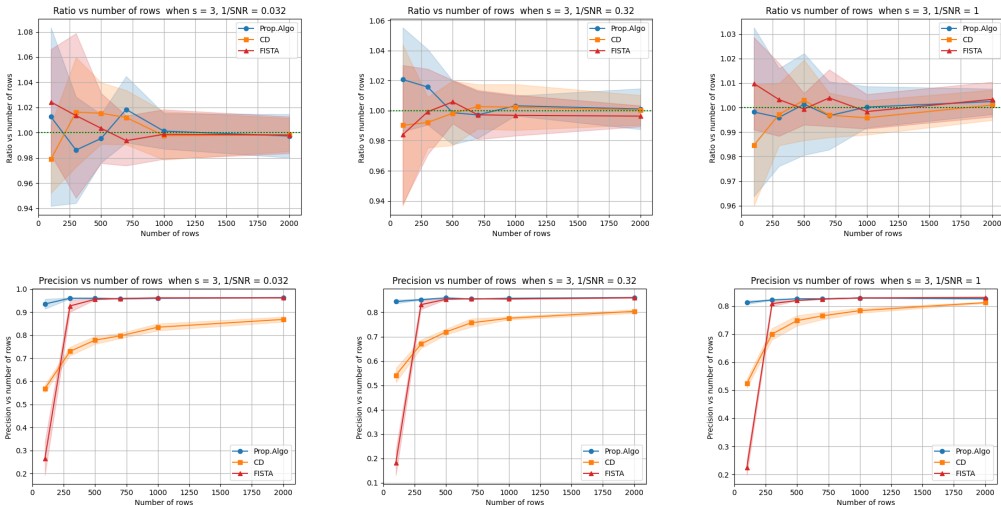

Figure 1: **Numerical results on synthetic data.** In short, each dot in the figure represents the average value of 10 *independent trials* (i.e., experiments) of compressed matrices $\boldsymbol{\Phi}^{(1)}, \dots, \boldsymbol{\Phi}^{(10)}$ on a given tuple of parameters $(K, d, n, \mathrm{SNR}, m)$. The shaded parts represent the empirical standard deviations over 10 trials. In the first row, we plot the ratio of training loss after and before compression, i.e., $\|\boldsymbol{\Phi Y} - \widehat{\boldsymbol{W}}\boldsymbol{X}\|_F^2 / \|\boldsymbol{Y} - \widehat{\boldsymbol{Z}}\boldsymbol{X}\|_F^2$ versus the number of rows $m$. It is obvious that the ratio converges to one as $m$ increases, which validates the result presented in Theorem 1. In the second row, we plot percision@3 versus the number of rows. As we can observe, the proposed algorithm outperforms CD and FISTA.

Based on Figure 1, we observe that the proposed algorithm enjoys a better computational cost and accuracy on most metrics. The running time for the proposed algorithm and baselines are reported in Table 2 (see in Appendix A.7), which further demonstrates the efficiency of the proposed algorithm. The implemented code could be found on Github `https://github.com/from-ryan/Solving_SHORE_via_compression`.

## 5 Conclusion and Future Directions

In conclusion, we propose a two-stage framework to solve Sparse & High-dimensional-Output REgression (SHORE) problem, the computational and statistical results indicate that the proposed framework is computationally scalable, maintaining the same order of both the training loss and prediction loss before and after compression under relatively weak sample set conditions, especially in the sparse and high-dimensional-output setting where the input dimension is polynomially smaller compared to the output dimension. In numerical experiments, SHORE provides improved optimization performance over existing MOR methods, for both synthetic data and real data.

We close with some potential questions for future investigation. The first is to extend our theoretical results to nonlinear/nonconvex SHORE frameworks [24]. The second direction is to improve existing variable reduction methods for better scalability while maintaining small sacrificing on prediction accuracy, e.g., new design and analysis on randomized projection matrices. The third direction is to explore general scenarios when high dimensional outputs enjoys additional geometric structures [30] from real applications in machine learning or operations management other than $s$-sparsity and its variants as discussed in the paper. Taking our result for SHORE as an initial start, we expect a stronger follow-up work that applies to MOR with additional structures, which eventually benefits the learning community in both practice and theory.

## Acknowledgement

Renyuan Li and Guanyi Wang were supported by the National University of Singapore under AcRF Tier-1 grant (A-8000607-00-00) 22-5539-A0001. Zhehui Chen would like to thank Google for its support in providing the research environment and supportive community that made this work possible.

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

# A   Appendix / supplemental material

## A.1   Proof of Theorem 1

Recall the following theorem in the main text.

**Theorem 1.**   For any $\delta \in (0, 1)$ and $\tau \in (0, 1)$, suppose the compressed matrix $\boldsymbol{\Phi}$ follows Assumption 1 with $m \geq O(\frac{1}{\delta^2} \cdot \log(\frac{K}{\tau}))$. We have the following inequality for training loss

$$\|\boldsymbol{\Phi Y} - \widehat{\boldsymbol{W}} \boldsymbol{X}\|_F^2 \leq (1 + \delta) \cdot \|\boldsymbol{Y} - \widehat{\boldsymbol{Z}} \boldsymbol{X}\|_F^2$$

holds with probability at least $1 - \tau$, where $\widehat{\boldsymbol{Z}}, \widehat{\boldsymbol{W}}$ are optimal solutions for the uncompressed (1) and compressed SHORE (2), respectively.

*Proof.*   Given a set of $n$ samples $\{(\boldsymbol{x}^i, \boldsymbol{y}^i)\}_{i=1}^n$, and a matrix $\boldsymbol{\Phi}$ generated from Assumption 1, we have

$$\|\boldsymbol{\Phi Y} - \widehat{\boldsymbol{W}} \boldsymbol{X}\|_F^2 \leq \|\boldsymbol{\Phi Y} - \boldsymbol{\Phi} \widehat{\boldsymbol{Z}} \boldsymbol{X}\|_F^2 = \|\boldsymbol{\Phi}(\boldsymbol{Y} - \widehat{\boldsymbol{Z}} \boldsymbol{X})\|_F^2 \; ,$$

where the inequality holds due to the optimality of $\widehat{\boldsymbol{W}}$. Let $\boldsymbol{Y}' = \boldsymbol{Y} - \widehat{\boldsymbol{Z}} \boldsymbol{X}$. Consider the singular value decomposition: $\boldsymbol{Y}' = \boldsymbol{U}_{\boldsymbol{Y}'} \boldsymbol{\Sigma}_{\boldsymbol{Y}'} \boldsymbol{V}_{\boldsymbol{Y}'}^\top$, and then we have

$$\|\boldsymbol{\Phi}(\boldsymbol{Y} - \widehat{\boldsymbol{Z}} \boldsymbol{X})\|_F^2 = \|\boldsymbol{\Phi} \boldsymbol{Y}'\|_F^2 = \|\boldsymbol{\Phi} \boldsymbol{U}_{\boldsymbol{Y}'} \boldsymbol{\Sigma}_{\boldsymbol{Y}'} \boldsymbol{V}_{\boldsymbol{Y}'}^\top\|_F^2 = \|\boldsymbol{\Phi} \boldsymbol{U}_{\boldsymbol{Y}'} \boldsymbol{\Sigma}_{\boldsymbol{Y}'}\|_F^2.$$

Set $\tilde{\boldsymbol{\Phi}} := \boldsymbol{\Phi} \boldsymbol{U}_{\boldsymbol{Y}'}$. Since the generalization method for $\boldsymbol{\Phi}$ ensures its $(1, \delta)$-RIP with probability $1 - \tau$ (from the Johnson-Lindenstrauss Lemma), and $\boldsymbol{U}_{\boldsymbol{Y}'}$ is a real unitary matrix, then Lemma 1 for *unitary invariant* shows that $\tilde{\boldsymbol{\Phi}}$ is also $(1, \delta)$-RIP with probability $1 - \tau$. Now, using $\tilde{\boldsymbol{\Phi}}$ is $(1, \delta)$-RIP and all columns in $\boldsymbol{\Sigma}_{\boldsymbol{Y}'}$ has at most one non-zero component, we have

$$\|\boldsymbol{\Phi} \boldsymbol{U}_{\boldsymbol{Y}'} \boldsymbol{\Sigma}_{\boldsymbol{Y}'}\|_F^2 = \|\tilde{\boldsymbol{\Phi}} \boldsymbol{\Sigma}_{\boldsymbol{Y}'}\|_F^2 \leq (1 + \delta) \|\boldsymbol{\Sigma}_{\boldsymbol{Y}'}\|_F^2 = (1 + \delta) \|\boldsymbol{Y}'\|_F^2.$$

Combining the above inequalities together implies

$$\|\boldsymbol{\Phi Y} - \widehat{\boldsymbol{W}} \boldsymbol{X}\|_F^2 \leq (1 + \delta) \|\boldsymbol{Y}'\|_F^2 = (1 + \delta) \cdot \|\boldsymbol{Y} - \widehat{\boldsymbol{Z}} \boldsymbol{X}\|_F^2,$$

which completes the proof. $\qquad\qquad\square$

## A.2   Proof of Theorem 2

Recall the following theorem in the main text.

**Theorem 2.**   For any $\delta \in (0, 1)$ and $\tau \in (0, 1)$, suppose the compressed matrix $\boldsymbol{\Phi}$ follows Assumption 1 with $m \geq O(\frac{s}{\delta^2} \log(\frac{K}{\tau}))$. With a fixed stepsize $\eta \in (\frac{1}{2-2\delta}, 1)$, the following inequality

$$\|\widehat{\boldsymbol{y}} - \boldsymbol{v}^{(t)}\|_2 \leq c_1^t \cdot \|\widehat{\boldsymbol{y}} - \boldsymbol{v}^{(0)}\|_2 + \frac{c_2}{1 - c_1} \cdot \|\boldsymbol{\Phi} \widehat{\boldsymbol{y}} - \widehat{\boldsymbol{W}} \boldsymbol{x}\|_2$$

holds for all $t \in [T]$ simultaneously with probability at least $1 - \tau$, where $c_1 := 2 - 2\eta + 2\eta\delta < 1$ is some positive constant strictly smaller than 1, and $c_2 := 2\eta\sqrt{1 + \delta}$ is some constant.

*Proof.*   Suppose Assumption 1 holds, then the randomized compressed matrix $\boldsymbol{\Phi}$ ensures $(3s, \delta)$-RIP with probability at least $1 - \tau$ (see Remark 3). Thus, to complete the proof, it is sufficient to show the above inequality holds for all $t \in [T]$ under such a compressed matrix $\boldsymbol{\Phi}$ with $(3s, \delta)$-RIP. We conclude that the proof is, in general, separated into three steps.

**Step-1.**   *We establish an upper bound on the $\ell_2$-norm distance between the current point and the optimal solution.* Due to the optimality of the projection (step-4 in Algorithm 1), we have the following inequality

$$\|\widetilde{\boldsymbol{v}}^{(t+1)} - \boldsymbol{v}^{(t+1)}\|_2^2 \leq \|\widetilde{\boldsymbol{v}}^{(t+1)} - \boldsymbol{v}\|_2^2$$

holds for all $\boldsymbol{v} \in \mathcal{V}_s^K \cap \mathcal{F}$, which further implies that

$$\|\widetilde{\boldsymbol{v}}^{(t+1)} - \boldsymbol{v} + \boldsymbol{v} - \boldsymbol{v}^{(t+1)}\|_2^2 \leq \|\widetilde{\boldsymbol{v}}^{(t+1)} - \boldsymbol{v}\|_2^2$$

$$\Leftrightarrow \|\boldsymbol{v} - \boldsymbol{v}^{(t+1)}\|_2^2 \leq 2\langle \boldsymbol{v} - \widetilde{\boldsymbol{v}}^{(t+1)}, \boldsymbol{v} - \boldsymbol{v}^{(t+1)}\rangle$$

holds for all $\boldsymbol{v} \in \mathcal{V}_s^K \cap \mathcal{F}$.

**Step-2.** *We show one-iteration improvement based on the above inequality.* Since $\widehat{\boldsymbol{y}} \in \mathcal{V}_s^K \cap \mathcal{F}$, we can replace $\boldsymbol{v}$ by $\widehat{\boldsymbol{y}}$, which still ensures the above inequality. Set $\boldsymbol{\Delta}^{(t)} := \widehat{\boldsymbol{y}} - \boldsymbol{v}^{(t)}$ for all $t \in [T]$. Based on the updating rule (step-3 in Algorithm 1), $\widetilde{\boldsymbol{v}}^{(t+1)} = \boldsymbol{v}^{(t)} - \eta \cdot \boldsymbol{\Phi}^\top (\boldsymbol{\Phi} \boldsymbol{v}^{(t)} - \widehat{\boldsymbol{W}} \boldsymbol{x})$. Thus, the above inequality can be written as

$$
\begin{aligned}
\|\boldsymbol{\Delta}^{(t+1)}\|_2^2 &\leq 2\langle \widehat{\boldsymbol{y}} - \boldsymbol{v}^{(t)} + \eta \cdot \boldsymbol{\Phi}^\top (\boldsymbol{\Phi} \boldsymbol{v}^{(t)} - \widehat{\boldsymbol{W}} \boldsymbol{x}), \boldsymbol{\Delta}^{(t+1)} \rangle \\
&= 2\langle \boldsymbol{\Delta}^{(t)}, \boldsymbol{\Delta}^{(t+1)} \rangle + 2\eta \langle \boldsymbol{\Phi} \boldsymbol{v}^{(t)} - \widehat{\boldsymbol{W}} \boldsymbol{x}, \boldsymbol{\Phi} \boldsymbol{\Delta}^{(t+1)} \rangle \\
&= 2\langle \boldsymbol{\Delta}^{(t)}, \boldsymbol{\Delta}^{(t+1)} \rangle - 2\eta \langle \boldsymbol{\Phi} \boldsymbol{\Delta}^{(t)}, \boldsymbol{\Phi} \boldsymbol{\Delta}^{(t+1)} \rangle + 2\eta \langle \boldsymbol{\Phi} \widehat{\boldsymbol{y}} - \widehat{\boldsymbol{W}} \boldsymbol{x}, \boldsymbol{\Phi} \boldsymbol{\Delta}^{(t+1)} \rangle. \quad (6)
\end{aligned}
$$

Using Lemma 2 with

$$
\frac{\boldsymbol{\Delta}^{(t)}}{\|\boldsymbol{\Delta}^{(t)}\|_2}, \quad \frac{\boldsymbol{\Delta}^{(t+1)}}{\|\boldsymbol{\Delta}^{(t+1)}\|_2}, \quad \frac{\boldsymbol{\Delta}^{(t)}}{\|\boldsymbol{\Delta}^{(t)}\|_2} + \frac{\boldsymbol{\Delta}^{(t+1)}}{\|\boldsymbol{\Delta}^{(t+1)}\|_2}, \quad \frac{\boldsymbol{\Delta}^{(t)}}{\|\boldsymbol{\Delta}^{(t)}\|_2} - \frac{\boldsymbol{\Delta}^{(t+1)}}{\|\boldsymbol{\Delta}^{(t+1)}\|_2}
$$

all $3s$-sparse vectors, and $\boldsymbol{\Phi}$ a $(3s, \delta)$-RIP matrix, we have

$$
-2\eta \left\langle \boldsymbol{\Phi} \frac{\boldsymbol{\Delta}^{(t)}}{\|\boldsymbol{\Delta}^{(t)}\|_2}, \boldsymbol{\Phi} \frac{\boldsymbol{\Delta}^{(t+1)}}{\|\boldsymbol{\Delta}^{(t+1)}\|_2} \right\rangle \leq 2\delta\eta - 2\eta \left\langle \frac{\boldsymbol{\Delta}^{(t)}}{\|\boldsymbol{\Delta}^{(t)}\|_2}, \frac{\boldsymbol{\Delta}^{(t+1)}}{\|\boldsymbol{\Delta}^{(t+1)}\|_2} \right\rangle,
$$

which implies

$$
-2\eta \langle \boldsymbol{\Phi} \boldsymbol{\Delta}^{(t)}, \boldsymbol{\Phi} \boldsymbol{\Delta}^{(t+1)} \rangle \leq 2\delta\eta \|\boldsymbol{\Delta}^{(t)}\|_2 \|\boldsymbol{\Delta}^{(t+1)}\|_2 - 2\eta \langle \boldsymbol{\Delta}^{(t)}, \boldsymbol{\Delta}^{(t+1)} \rangle.
$$

Inserting the above result into inequality (6) gives

$$
\begin{aligned}
\|\boldsymbol{\Delta}^{(t+1)}\|_2^2 &\leq (2 - 2\eta)\langle \boldsymbol{\Delta}^{(t)}, \boldsymbol{\Delta}^{(t+1)} \rangle + 2\delta\eta \|\boldsymbol{\Delta}^{(t)}\|_2 \|\boldsymbol{\Delta}^{(t+1)}\|_2 + 2\eta \langle \boldsymbol{\Phi} \widehat{\boldsymbol{y}} - \widehat{\boldsymbol{W}} \boldsymbol{x}, \boldsymbol{\Phi} \boldsymbol{\Delta}^{(t+1)} \rangle \\
&\overset{\text{(i)}}{\leq} (2 - 2\eta + 2\eta\delta)\|\boldsymbol{\Delta}^{(t)}\|_2 \|\boldsymbol{\Delta}^{(t+1)}\|_2 + 2\eta \|\boldsymbol{\Phi} \widehat{\boldsymbol{y}} - \widehat{\boldsymbol{W}} \boldsymbol{x}\|_2 \|\boldsymbol{\Phi} \boldsymbol{\Delta}^{(t+1)}\|_2 \\
&\leq (2 - 2\eta + 2\eta\delta)\|\boldsymbol{\Delta}^{(t)}\|_2 \|\boldsymbol{\Delta}^{(t+1)}\|_2 + 2\eta\sqrt{1+\delta} \|\boldsymbol{\Phi} \widehat{\boldsymbol{y}} - \widehat{\boldsymbol{W}} \boldsymbol{x}\|_2 \|\boldsymbol{\Delta}^{(t+1)}\|_2,
\end{aligned}
$$

where the above inequality (i) requests $\eta < 1$ to ensure the inequality $(2 - 2\eta)\langle \boldsymbol{\Delta}^{(t)}, \boldsymbol{\Delta}^{(t+1)} \rangle \leq (2 - 2\eta)\|\boldsymbol{\Delta}^{(t)}\|_2 \|\boldsymbol{\Delta}^{(t+1)}\|_2$ holds. Therefore, dividing $\|\boldsymbol{\Delta}^{(t+1)}\|_2$ on both side implies

$$
\|\boldsymbol{\Delta}^{(t+1)}\|_2 \leq (2 - 2\eta + 2\eta\delta)\|\boldsymbol{\Delta}^{(t)}\|_2 + 2\eta\sqrt{1+\delta} \|\boldsymbol{\Phi} \widehat{\boldsymbol{y}} - \widehat{\boldsymbol{W}} \boldsymbol{x}\|_2, \quad (7)
$$

which gives the one-step improvement.

**Step-3.** *Combine everything together.* To ensure contractions in every iteration, we pick stepsize $\eta$ such that $2 - 2\eta + 2\eta\delta \in (0, 1)$ with $\eta < 1$, which gives $\eta \in \left( \frac{1}{2(1-\delta)}, 1 \right)$. Using the above inequality (7) for one-step improvement, we have

$$
\|\widehat{\boldsymbol{y}} - \boldsymbol{v}^{(t)}\|_2 \leq (2 - 2\eta + 2\eta\delta)^t \cdot \|\widehat{\boldsymbol{y}} - \boldsymbol{v}^{(0)}\|_2 + \frac{2\eta\sqrt{1+\delta}}{2\eta - 2\eta\delta - 1} \|\boldsymbol{\Phi} \widehat{\boldsymbol{y}} - \widehat{\boldsymbol{W}} \boldsymbol{x}\|_2,
$$

which completes the proof. $\qquad\square$

### A.3 Proof of Theorem 3

Recall the following theorem in the main text.

**Theorem 3.** For any $\delta \in (0, 1)$ and $\tau \in (0, \frac{1}{3})$, suppose compressed matrix $\boldsymbol{\Phi}$ follows Assumption 1 with $m \geq O(\frac{s}{\delta^2} \log(\frac{K}{\tau}))$, and Assumption 2 holds, for any constant $\epsilon > 0$, the following results hold:

(Matrix Error). The inequality for matrix error $\left\| \boldsymbol{M}_{\boldsymbol{xx}}^{1/2} \widehat{\boldsymbol{M}}_{\boldsymbol{xx}}^{-1} \boldsymbol{M}_{\boldsymbol{xx}}^{1/2} \right\|_{\text{op}} \leq 4$ holds with probability at least $1 - 2\tau$ as the number of samples $n \geq n_1$ with

$$
n_1 := \max \left\{ \frac{64C^2\sigma^4}{9\lambda_{\min}^2(\boldsymbol{M}_{\boldsymbol{xx}})} (d + \log(2/\tau)), \frac{32^2 \|\boldsymbol{\mu}_{\boldsymbol{x}}\|_2^2 \sigma^2}{\lambda_{\min}^2(\boldsymbol{M}_{\boldsymbol{xx}})} \left( 2\sqrt{d} + \sqrt{\log(1/\tau)} \right)^2 \right\},
$$

where $C$ is some fixed positive constant used in matrix concentration inequality of operator norm.

(Uncompressed). The generalization error bound for uncompressed SHORE satisfies $\mathcal{L}(\widehat{\boldsymbol{Z}}) \leq \mathcal{L}(\boldsymbol{Z}_*) + 4\epsilon$ with probability at least $1 - 3\tau$, as the number of samples $n \geq \max\{n_1, n_2\}$ with

$$
n_2 := \max\left\{ 4(\|\boldsymbol{Z}_*\|_F^2 + K) \cdot \frac{d + 2\sqrt{d\log(K/\tau)} + 2\log(K/\tau)}{\epsilon}, \ 4\|\boldsymbol{\mu_y} - \boldsymbol{Z}_*\boldsymbol{\mu_x}\|_2^2 \cdot \frac{d}{\epsilon} \right\}.
$$

(Compressed). The generalization error bound for the compressed SHORE satisfies $\mathcal{L}^{\boldsymbol{\Phi}}(\widehat{\boldsymbol{W}}) \leq \mathcal{L}^{\boldsymbol{\Phi}}(\boldsymbol{W}_*) + 4\epsilon$ with probability at least $1 - 3\tau$, as the number of sample $n \geq \max\{n_1, \widetilde{n}_2\}$ with

$$
\widetilde{n}_2 := \max\left\{ 4(\|\boldsymbol{W}_*\|_F^2 + \|\boldsymbol{\Phi}\|_F^2) \cdot \frac{d + 2\sqrt{d\log(m/\tau)} + 2\log(m/\tau)}{\epsilon}, \ 4\|\boldsymbol{\Phi}\boldsymbol{\mu_y} - \boldsymbol{W}_*\boldsymbol{\mu_x}\|_2^2 \cdot \frac{d}{\epsilon} \right\}.
$$

*Proof.* Let us start with the uncompressed version.

**Step-1.** Note that the optimal solutions for population loss and empirical loss are

$$
\boldsymbol{Z}_* = \boldsymbol{M_{yx}}\boldsymbol{M_{xx}}^{-1} \quad \text{and} \quad \widehat{\boldsymbol{Z}} = \frac{\boldsymbol{Y}\boldsymbol{X}^\top}{n}\left(\frac{\boldsymbol{X}\boldsymbol{X}^\top}{n}\right)^{-1} =: \widehat{\boldsymbol{M}}_{\boldsymbol{yx}}\widehat{\boldsymbol{M}}_{\boldsymbol{xx}}^{-1},
$$

respectively. Thus, the generalization error bound is

$$
\mathcal{L}(\widehat{\boldsymbol{Z}}) - \mathcal{L}(\boldsymbol{Z}_*) = \|\widehat{\boldsymbol{Z}} - \boldsymbol{Z}_*\|_{\boldsymbol{M_{xx}}}^2 = \|(\widehat{\boldsymbol{Z}} - \boldsymbol{Z}_*)\boldsymbol{M}_{\boldsymbol{xx}}^{1/2}\|_F^2.
$$

Note that

$$
\begin{aligned}
(\widehat{\boldsymbol{Z}} - \boldsymbol{Z}_*)\boldsymbol{M}_{\boldsymbol{xx}}^{1/2} &= \left(\widehat{\boldsymbol{M}}_{\boldsymbol{yx}}\widehat{\boldsymbol{M}}_{\boldsymbol{xx}}^{-1} - \boldsymbol{M_{yx}}\boldsymbol{M_{xx}}^{-1}\right)\boldsymbol{M}_{\boldsymbol{xx}}^{1/2} \\
&= \left(\widehat{\boldsymbol{M}}_{\boldsymbol{yx}} - \boldsymbol{M_{yx}}\boldsymbol{M_{xx}}^{-1}\widehat{\boldsymbol{M}}_{\boldsymbol{xx}}\right)\widehat{\boldsymbol{M}}_{\boldsymbol{xx}}^{-1}\boldsymbol{M}_{\boldsymbol{xx}}^{1/2} \\
&= \widehat{\mathbb{E}}[\boldsymbol{y}\boldsymbol{x}^\top - \boldsymbol{Z}_*\boldsymbol{x}\boldsymbol{x}^\top]\widehat{\boldsymbol{M}}_{\boldsymbol{xx}}^{-1}\boldsymbol{M}_{\boldsymbol{xx}}^{1/2} \\
&= \widehat{\mathbb{E}}[(\boldsymbol{y} - \boldsymbol{Z}_*\boldsymbol{x})\boldsymbol{x}^\top\widehat{\boldsymbol{M}}_{\boldsymbol{xx}}^{-1/2}]\widehat{\boldsymbol{M}}_{\boldsymbol{xx}}^{-1/2}\boldsymbol{M}_{\boldsymbol{xx}}^{1/2},
\end{aligned}
$$

where we use $\widehat{\mathbb{E}}[\cdot]$ to denote the empirical distribution. Then, the above generalization error bound can be upper-bounded as follows

$$
\begin{aligned}
\left\|(\widehat{\boldsymbol{Z}} - \boldsymbol{Z}_*)\boldsymbol{M}_{\boldsymbol{xx}}^{1/2}\right\|_F &= \left\|\widehat{\mathbb{E}}[(\boldsymbol{y} - \boldsymbol{Z}_*\boldsymbol{x})\boldsymbol{x}^\top\widehat{\boldsymbol{M}}_{\boldsymbol{xx}}^{-1/2}]\widehat{\boldsymbol{M}}_{\boldsymbol{xx}}^{-1/2}\boldsymbol{M}_{\boldsymbol{xx}}^{1/2}\right\|_F \\
&\leq \underbrace{\left\|\widehat{\mathbb{E}}[(\boldsymbol{y} - \boldsymbol{Z}_*\boldsymbol{x})\boldsymbol{x}^\top\widehat{\boldsymbol{M}}_{\boldsymbol{xx}}^{-1/2}]\right\|_F}_{\text{rescaled approximation error}} \underbrace{\left\|\widehat{\boldsymbol{M}}_{\boldsymbol{xx}}^{-1/2}\boldsymbol{M}_{\boldsymbol{xx}}^{1/2}\right\|_{\text{op}}}_{\text{matrix error}}.
\end{aligned}
$$

**Step-2.** Next, we provide upper bounds on these two terms $\left\|\widehat{\mathbb{E}}[(\boldsymbol{y} - \boldsymbol{Z}_*\boldsymbol{x})\boldsymbol{x}^\top\widehat{\boldsymbol{M}}_{\boldsymbol{xx}}^{-1/2}]\right\|_F$ and $\left\|\widehat{\boldsymbol{M}}_{\boldsymbol{xx}}^{-1/2}\boldsymbol{M}_{\boldsymbol{xx}}^{1/2}\right\|_{\text{op}}$ in the right-hand-side separately.

*For matrix error term* $\left\|\widehat{\boldsymbol{M}}_{\boldsymbol{xx}}^{-1/2}\boldsymbol{M}_{\boldsymbol{xx}}^{1/2}\right\|_{op}$, we have

$$
\left\|\widehat{\boldsymbol{M}}_{\boldsymbol{xx}}^{-1/2}\boldsymbol{M}_{\boldsymbol{xx}}^{1/2}\right\|_{\text{op}}^2 = \left\|\boldsymbol{M}_{\boldsymbol{xx}}^{1/2}\widehat{\boldsymbol{M}}_{\boldsymbol{xx}}^{-1}\boldsymbol{M}_{\boldsymbol{xx}}^{1/2}\right\|_{\text{op}}.
$$

Due to Assumption 2, the centralized feature vector $\boldsymbol{\xi_x} := \boldsymbol{x} - \boldsymbol{\mu_x}$ ensures the following inequality

$$
\mathbb{E}_{\mathcal{D}}\left[\exp\left(\lambda\boldsymbol{v}^\top\boldsymbol{\xi_x}\right)\right] \leq \exp\left(\frac{\lambda^2\|\boldsymbol{v}\|_2^2\sigma^2}{2}\right)
$$

for all $\boldsymbol{v} \in \mathbb{R}^d$. Consider the empirical second moment of $\boldsymbol{x}$,

$$
\widehat{\boldsymbol{M}}_{\boldsymbol{xx}} = \sum_{i=1}^n \frac{\boldsymbol{x}^i(\boldsymbol{x}^i)^\top}{n} = \sum_{i=1}^n \frac{\boldsymbol{\xi}_{\boldsymbol{x}}^i(\boldsymbol{\xi}_{\boldsymbol{x}}^i)^\top}{n} + \boldsymbol{\mu_x}\left(\sum_{i=1}^n \frac{\boldsymbol{\xi}_{\boldsymbol{x}}^i}{n}\right)^\top + \left(\sum_{i=1}^n \frac{\boldsymbol{\xi}_{\boldsymbol{x}}^i}{n}\right)\boldsymbol{\mu_x}^\top + \boldsymbol{\mu_x}\boldsymbol{\mu_x}^\top.
$$

Since $\boldsymbol{\xi}_x^1, \ldots, \boldsymbol{\xi}_x^n$ are i.i.d. $\sigma^2$-subGaussian random vector with zero mean and covariance matrix $\boldsymbol{\Sigma}_{xx}$, then based on Lemma 3, there exists a positive constant $C$ such that for any $\tau \in (0, 1)$,

$$\mathbb{P}_{\mathcal{D}} \left( \left\| \sum_{i=1}^n \frac{\boldsymbol{\xi}_x^i (\boldsymbol{\xi}_x^i)^\top}{n} - \boldsymbol{\Sigma}_{xx} \right\|_{\mathrm{op}} \leq C\sigma^2 \max \left\{ \sqrt{\frac{d + \log(2/\tau)}{n}}, \frac{d + \log(2/\tau)}{n} \right\} \right) \geq 1 - \tau,$$

and based on Lemma 4, for any $\tau \in (0, 1)$,

$$\mathbb{P}_{\mathcal{D}} \left( \left\| \sum_{i=1}^n \frac{\boldsymbol{\xi}_x^i}{n} \right\|_2 \leq \frac{4\sigma\sqrt{d} + 2\sigma\sqrt{\log(1/\tau)}}{\sqrt{n}} \right) \geq 1 - \tau.$$

Let $\boldsymbol{\Delta}_{xx} := \sum_{i=1}^n \frac{\boldsymbol{\xi}_x^i (\boldsymbol{\xi}_x^i)^\top}{n} - \boldsymbol{\Sigma}_{xx}$ and $\bar{\boldsymbol{\xi}} := \sum_{i=1}^n \frac{\boldsymbol{\xi}_x^i}{n}$, then $\widehat{\boldsymbol{M}}_{xx}$ can be represented by

$$\widehat{\boldsymbol{M}}_{xx} = \underbrace{\boldsymbol{\Sigma}_{xx} + \boldsymbol{\mu}_x \boldsymbol{\mu}_x^\top}_{=: \boldsymbol{M}_{xx}} + \boldsymbol{\Delta}_{xx} + \boldsymbol{\mu}_x \bar{\boldsymbol{\xi}}^\top + \bar{\boldsymbol{\xi}} \boldsymbol{\mu}_x^\top,$$

and thus we have $\boldsymbol{M}_{xx}^{-1/2} \widehat{\boldsymbol{M}}_{xx} \boldsymbol{M}_{xx}^{-1/2} = \boldsymbol{I}_d + \boldsymbol{M}_{xx}^{-1/2} \left( \boldsymbol{\Delta}_{xx} + \boldsymbol{\mu}_x \bar{\boldsymbol{\xi}}^\top + \bar{\boldsymbol{\xi}} \boldsymbol{\mu}_x^\top \right) \boldsymbol{M}_{xx}^{-1/2}$. Then the minimum eigenvalue of $\boldsymbol{M}_{xx}^{-1/2} \widehat{\boldsymbol{M}}_{xx} \boldsymbol{M}_{xx}^{-1/2}$ can be lower bounded as follows

$$\lambda_{\min} \left( \boldsymbol{M}_{xx}^{-1/2} \widehat{\boldsymbol{M}}_{xx} \boldsymbol{M}_{xx}^{-1/2} \right)$$

$$\geq 1 - \left\| \boldsymbol{M}_{xx}^{-1/2} \left( \boldsymbol{\Delta}_{xx} + \boldsymbol{\mu}_x \bar{\boldsymbol{\xi}}^\top + \bar{\boldsymbol{\xi}} \boldsymbol{\mu}_x^\top \right) \boldsymbol{M}_{xx}^{-1/2} \right\|_{\mathrm{op}}$$

$$\geq 1 - \left\| \boldsymbol{M}_{xx}^{-1/2} \boldsymbol{\Delta}_{xx} \boldsymbol{M}_{xx}^{-1/2} \right\|_{\mathrm{op}} - \left\| \boldsymbol{M}_{xx}^{-1/2} \left( \boldsymbol{\mu}_x \bar{\boldsymbol{\xi}}^\top + \bar{\boldsymbol{\xi}} \boldsymbol{\mu}_x^\top \right) \boldsymbol{M}_{xx}^{-1/2} \right\|_{\mathrm{op}}$$

$$\geq 1 - \frac{C\sigma^2}{\lambda_{\min}(\boldsymbol{M}_{xx})} \sqrt{\frac{d + \log(2/\tau)}{n}} - \frac{2\|\boldsymbol{\mu}_x\|_2}{\lambda_{\min}(\boldsymbol{M}_{xx})} \frac{4\sigma\sqrt{d} + 2\sigma\sqrt{\log(1/\tau)}}{\sqrt{n}},$$

where the final inequality holds with probability at least $1 - 2\tau$ by inserting the above non-asymptotic bounds. Then, we have

$$\left\| \boldsymbol{M}_{xx}^{1/2} \widehat{\boldsymbol{M}}_{xx}^{-1} \boldsymbol{M}_{xx}^{1/2} \right\|_{\mathrm{op}}$$

$$= \lambda_{\min}^{-1} \left( \boldsymbol{M}_{xx}^{-1/2} \widehat{\boldsymbol{M}}_{xx} \boldsymbol{M}_{xx}^{-1/2} \right)$$

$$\leq \left[ 1 - \frac{C\sigma^2}{\lambda_{\min}(\boldsymbol{M}_{xx})} \sqrt{\frac{d + \log(2/\tau)}{n}} - \frac{2\|\boldsymbol{\mu}_x\|_2}{\lambda_{\min}(\boldsymbol{M}_{xx})} \frac{4\sigma\sqrt{d} + 2\sigma\sqrt{\log(1/\tau)}}{\sqrt{n}} \right]^{-1}$$

holds with probability at least $1 - 2\tau$. It is easy to observe that as

$$n \geq n_1 := \max \left\{ \frac{64C^2\sigma^4}{9\lambda_{\min}^2(\boldsymbol{M}_{xx})} (d + \log(2/\tau)), \frac{32^2 \|\boldsymbol{\mu}_x\|_2^2 \sigma^2}{\lambda_{\min}^2(\boldsymbol{M}_{xx})} \left( 2\sqrt{d} + \sqrt{\log(1/\tau)} \right)^2 \right\},$$

we have $\left\| \boldsymbol{M}_{xx}^{1/2} \widehat{\boldsymbol{M}}_{xx}^{-1} \boldsymbol{M}_{xx}^{1/2} \right\|_{\mathrm{op}} \leq 4$ holds with probability $1 - 2\tau$.

*For rescaled approximation error term* $\left\| \widehat{\mathbb{E}}[(\boldsymbol{y} - \boldsymbol{Z}_* \boldsymbol{x}) \boldsymbol{x}^\top \widehat{\boldsymbol{M}}_{xx}^{-1/2}] \right\|_F$, we first compute variance proxy for the subGaussian vector $\boldsymbol{y} - \boldsymbol{Z}_* \boldsymbol{x}$. Note that the $j$-th component of the subGaussian vector $\boldsymbol{y} - \boldsymbol{Z}_* \boldsymbol{x}$ can be written as

$$[\boldsymbol{y} - \boldsymbol{Z}_* \boldsymbol{x}]_j = (-[\boldsymbol{Z}_*]_{j,:}^\top \mid \boldsymbol{e}_j^\top) \begin{pmatrix} \boldsymbol{x} \\ \boldsymbol{y} \end{pmatrix},$$

where $[\boldsymbol{Z}_*]_{j,:}^\top$ is the $j$-th row of $\boldsymbol{Z}_*$, $\boldsymbol{e}_j^\top$ is a $K$-dimensional vector with $j$-th component equals to one and rest components equal to zero. Thus, it is easy to observe that the $\ell_2$-norm square of $(-[\boldsymbol{Z}_*]_{j,:}^\top \mid \boldsymbol{e}_j^\top)$ is $\|[\boldsymbol{Z}_*]_{j,:}\|_2^2 + 1$, and therefore, based on the Assumption 2, we have

$$\mathbb{E}_{\mathcal{D}} \left[ \exp \left( \lambda [\boldsymbol{y} - \boldsymbol{Z}_* \boldsymbol{x}]_j - \lambda [\boldsymbol{\mu}_y - \boldsymbol{Z}_* \boldsymbol{\mu}_x]_j \right) \right] \leq \exp \left( \lambda^2 \cdot (\|[\boldsymbol{Z}_*]_{j,:}\|_2^2 + 1)\sigma^2/2 \right),$$

i.e., a subGaussian with variance proxy $\sigma_j^2 := (\|[Z_*]_{j,:}\|_2^2 + 1)\sigma^2$. Thus the rescaled approximation error can be upper-bounded by

$$
\left\| \widehat{\mathbb{E}}[(y - Z_* x)x^\top \widehat{M}_{xx}^{-1/2}] \right\|_F^2
$$

$$
= \sum_{j=1}^K \left\| \widehat{\mathbb{E}}[[y - Z_* x]_j x^\top \widehat{M}_{xx}^{-1/2}] \right\|_2^2
$$

$$
\leq 2 \sum_{j=1}^K \underbrace{\left\| \widehat{\mathbb{E}}[([y - Z_* x]_j - [\mu_y - Z_* \mu_x]_j)x^\top \widehat{M}_{xx}^{-1/2}] \right\|_2^2}_{=:T_j^1} + 2 \sum_{j=1}^K \underbrace{\left\| \widehat{\mathbb{E}}[[\mu_y - Z_* \mu_x]_j x^\top \widehat{M}_{xx}^{-1/2}] \right\|_2^2}_{=:T_j^2}.
$$

We control term $T_j^1$ for all $j \in [K]$ separately using Lemma 5 as follows: For all $\tau \in (0, 1)$, we have

$$
\mathbb{P}_{\mathcal{D}} \left( T_j^1 \leq \frac{\sigma_j^2 (d + 2\sqrt{d \log(K/\tau)} + 2 \log(K/\tau))}{n} \right) \geq 1 - \tau/K.
$$

For the term $T_j^2$, we have

$$
T_j^2 = \left\| \frac{1}{n} \sum_{i=1}^n [\mu_y - Z_* \mu_x]_j (x^i)^\top \widehat{M}_{xx}^{-1/2} \right\|_2^2
$$

$$
= \left\| \frac{[\mu_y - Z_* \mu_x]_j}{\sqrt{n}} \left( \widehat{M}_{xx}^{-1/2} \frac{x^1}{\sqrt{n}} \mid \cdots \mid \widehat{M}_{xx}^{-1/2} \frac{x^n}{\sqrt{n}} \right) 1_n \right\|_2^2
$$

$$
= \frac{[\mu_y - Z_* \mu_x]_j^2}{n} \left\| \widehat{M}_{xx}^{-1/2} \left( \frac{x^1}{\sqrt{n}} \mid \cdots \mid \frac{x^n}{\sqrt{n}} \right) 1_n \right\|_2^2.
$$

Now let $\left( \frac{x^1}{\sqrt{n}} \mid \cdots \mid \frac{x^n}{\sqrt{n}} \right) = U_x D_x V_x^\top$ be the singular value decomposition of the matrix $\left( \frac{x^1}{\sqrt{n}} \mid \cdots \mid \frac{x^n}{\sqrt{n}} \right)$, the above $\ell_2$-norm can be further written as

$$
\frac{[\mu_y - Z_* \mu_x]_j^2}{n} \left\| \widehat{M}_{xx}^{-1/2} \left( \frac{x^1}{\sqrt{n}} \mid \cdots \mid \frac{x^n}{\sqrt{n}} \right) 1_n \right\|_2^2
$$

$$
\overset{(\text{i})}{=} \frac{[\mu_y - Z_* \mu_x]_j^2}{n} 1_n^\top V_x \begin{pmatrix} I_d & 0_{d \times (n-d)} \\ 0_{(n-d) \times d} & 0_{(n-d) \times (n-d)} \end{pmatrix} V_x^\top 1_n
$$

$$
\overset{(\text{ii})}{=} \frac{[\mu_y - Z_* \mu_x]_j^2}{n} \cdot d,
$$

where the equality (i) holds due to the definition of empirical matrix $\widehat{M}_{xx} = \frac{1}{n} \sum_{i=1}^n x^i (x^i)^\top$, the equality (ii) holds due to the unitary property of matrix $V_x$. Combining the above two parts implies

$$
\left\| \widehat{\mathbb{E}}[(y - Z_* x)x^\top \widehat{M}_{xx}^{-1/2}] \right\|_F^2
$$

$$
= \sum_{j=1}^K \left\| \widehat{\mathbb{E}}[[y - Z_* x]_j x^\top \widehat{M}_{xx}^{-1/2}] \right\|_2^2
$$

$$
\leq 2 \sum_{j=1}^K T_j^1 + 2 \sum_{j=1}^K T_j^2
$$

$$
\overset{(\text{iii})}{\leq} 2 \sum_{j=1}^K \frac{\sigma_j^2 (d + 2\sqrt{d \log(K/\tau)} + 2 \log(K/\tau))}{n} + 2 \sum_{j=1}^K \frac{[\mu_y - Z_* \mu_x]_j^2}{n} \cdot d
$$

$$
= 2(\|Z_*\|_F^2 + K) \cdot \frac{d + 2\sqrt{d \log(K/\tau)} + 2 \log(K/\tau)}{n} + 2\|\mu_y - Z_* \mu_x\|_2^2 \cdot \frac{d}{n}
$$

with inequality (iii) holds with probability at least $1 - \tau$. Still, it is easy to observe that for any positive constant $\epsilon$, as

$$n \geq n_2 := \max\left\{ 4(\|\boldsymbol{Z}_*\|_F^2 + K) \cdot \frac{d + 2\sqrt{d\log(K/\tau)} + 2\log(K/\tau)}{\epsilon}, \ 4\|\boldsymbol{\mu_y} - \boldsymbol{Z}_*\boldsymbol{\mu_x}\|_2^2 \cdot \frac{d}{\epsilon} \right\},$$

we have $\left\|\widehat{\mathbb{E}}[(\boldsymbol{y} - \boldsymbol{Z}_*\boldsymbol{x})\boldsymbol{x}^\top \widehat{\boldsymbol{M}}_{\boldsymbol{xx}}^{-1/2}]\right\|_F^2 \leq \epsilon$ holds with probability at least $1 - \tau$.

**Step-3.** Combining two upper bounds together, if $n \geq \max\{n_1, n_2\}$, the following inequality for generalization error bound

$$\mathcal{L}(\widehat{\boldsymbol{Z}}) - \mathcal{L}(\boldsymbol{Z}_*) \leq \left\|\widehat{\mathbb{E}}[(\boldsymbol{y} - \boldsymbol{Z}_*\boldsymbol{x})\boldsymbol{x}^\top \widehat{\boldsymbol{M}}_{\boldsymbol{xx}}^{-1/2}]\right\|_F^2 \cdot \left\|\boldsymbol{M}_{\boldsymbol{xx}}^{1/2} \widehat{\boldsymbol{M}}_{\boldsymbol{xx}}^{-1} \boldsymbol{M}_{\boldsymbol{xx}}^{1/2}\right\|_{\mathrm{op}} \leq 4\epsilon$$

holds with probability at least $1 - 3\tau$.

*We then study the compressed version.*

**Step-1'.** Similarly, its optimal solutions for population loss and empirical loss are $\boldsymbol{W}_* = \boldsymbol{\Phi}\boldsymbol{M}_{\boldsymbol{yx}}\boldsymbol{M}_{\boldsymbol{xx}}^{-1}$ and $\widehat{\boldsymbol{W}} = \frac{\boldsymbol{\Phi Y X}^\top}{n}\left(\frac{\boldsymbol{XX}^\top}{n}\right)^{-1} =: \boldsymbol{\Phi}\widehat{\boldsymbol{M}}_{\boldsymbol{yx}}\widehat{\boldsymbol{M}}_{\boldsymbol{xx}}^{-1}$, respectively. Thus, the generalization error bound is

$$\mathcal{L}^{\boldsymbol{\Phi}}(\widehat{\boldsymbol{W}}) - \mathcal{L}^{\boldsymbol{\Phi}}(\boldsymbol{W}_*) = \|\widehat{\boldsymbol{W}} - \boldsymbol{W}_*\|_{\boldsymbol{M}_{\boldsymbol{xx}}}^2 = \|(\widehat{\boldsymbol{W}} - \boldsymbol{W}_*)\boldsymbol{M}_{\boldsymbol{xx}}^{1/2}\|_F^2.$$

Still, we have $(\widehat{\boldsymbol{W}} - \boldsymbol{W}_*)\boldsymbol{M}_{\boldsymbol{xx}}^{1/2} = \widehat{\mathbb{E}}[(\boldsymbol{\Phi}\boldsymbol{y} - \boldsymbol{W}_*\boldsymbol{x})\boldsymbol{x}^\top \widehat{\boldsymbol{M}}_{\boldsymbol{xx}}^{-1/2}]\widehat{\boldsymbol{M}}_{\boldsymbol{xx}}^{-1/2}\boldsymbol{M}_{\boldsymbol{xx}}^{1/2}$, and therefore, the generalization error bound can be upper-bounded by

$$\left\|(\widehat{\boldsymbol{W}} - \boldsymbol{W}_*)\boldsymbol{M}_{\boldsymbol{xx}}^{1/2}\right\|_F^2 \leq \left\|\widehat{\mathbb{E}}[(\boldsymbol{\Phi}\boldsymbol{y} - \boldsymbol{W}_*\boldsymbol{x})\boldsymbol{x}^\top \widehat{\boldsymbol{M}}_{\boldsymbol{xx}}^{-1/2}]\right\|_F^2 \left\|\widehat{\boldsymbol{M}}_{\boldsymbol{xx}}^{-1/2}\boldsymbol{M}_{\boldsymbol{xx}}^{1/2}\right\|_{\mathrm{op}}^2.$$

**Step-2'.** Next, we provide upper bounds on these two terms $\left\|\widehat{\mathbb{E}}[(\boldsymbol{\Phi}\boldsymbol{y} - \boldsymbol{W}_*\boldsymbol{x})\boldsymbol{x}^\top \widehat{\boldsymbol{M}}_{\boldsymbol{xx}}^{-1/2}]\right\|_F$ and $\left\|\widehat{\boldsymbol{M}}_{\boldsymbol{xx}}^{-1/2}\boldsymbol{M}_{\boldsymbol{xx}}^{1/2}\right\|_{\mathrm{op}}$ in the right-hand-side separately. Note that for the matrix error term $\left\|\widehat{\boldsymbol{M}}_{\boldsymbol{xx}}^{-1/2}\boldsymbol{M}_{\boldsymbol{xx}}^{1/2}\right\|_{\mathrm{op}}$, we could use the same upper bounded as mentioned in the proof of uncompressed version.

Now, to give the upper bound on the rescaled approximation error term $\left\|\widehat{\mathbb{E}}[(\boldsymbol{\Phi}\boldsymbol{y} - \boldsymbol{W}_*\boldsymbol{x})\boldsymbol{x}^\top \widehat{\boldsymbol{M}}_{\boldsymbol{xx}}^{-1/2}]\right\|_F$, we first compute the variance proxy for the subGaussian vector $\boldsymbol{\Phi}\boldsymbol{y} - \boldsymbol{W}_*\boldsymbol{x}$, which is

$$\widetilde{\sigma}_j^2 := (\|[\boldsymbol{W}_*]_{j,:}\|_2^2 + \|\boldsymbol{\Phi}_{j,:}\|_2^2)\sigma^2$$

for all $j \in [m]$. Thus, the rescaled approximation error for the compressed version can be upper bounded by

$$\left\|\widehat{\mathbb{E}}[(\boldsymbol{\Phi}\boldsymbol{y} - \boldsymbol{W}_*\boldsymbol{x})\boldsymbol{x}^\top \widehat{\boldsymbol{M}}_{\boldsymbol{xx}}^{-1/2}]\right\|_F^2 = \sum_{j=1}^m \left\|\widehat{\mathbb{E}}[[\boldsymbol{\Phi}\boldsymbol{y} - \boldsymbol{W}_*\boldsymbol{x}]_j \boldsymbol{x}^\top \widehat{\boldsymbol{M}}_{\boldsymbol{xx}}^{-1/2}]\right\|_2^2$$

$$\leq 2\sum_{j=1}^m \underbrace{\left\|\widehat{\mathbb{E}}[([\boldsymbol{\Phi}\boldsymbol{y} - \boldsymbol{W}_*\boldsymbol{x}]_j - [\boldsymbol{\Phi}\boldsymbol{\mu_y} - \boldsymbol{W}_*\boldsymbol{\mu_x}]_j)\boldsymbol{x}^\top \widehat{\boldsymbol{M}}_{\boldsymbol{xx}}^{-1/2}]\right\|_2^2}_{=:\widetilde{T}_j^1}$$

$$+ 2\sum_{j=1}^m \underbrace{\left\|\widehat{\mathbb{E}}[[\boldsymbol{\Phi}\boldsymbol{\mu_y} - \boldsymbol{W}_*\boldsymbol{\mu_x}]_j \boldsymbol{x}^\top \widehat{\boldsymbol{M}}_{\boldsymbol{xx}}^{-1/2}]\right\|_2^2}_{=:\widetilde{T}_j^2}.$$

Still using Lemma 5, for all $\tau \in (0, 1)$, we have

$$\mathbb{P}_{\mathcal{D}}\left(\widetilde{T}_j^1 \leq \frac{\widetilde{\sigma}_j^2(d + 2\sqrt{d\log(m/\tau)} + 2\log(m/\tau))}{n}\right) \geq 1 - \tau/m.$$

For the term $\widetilde{T}_j^2$, following the same proof procedures of the uncompressed version implies

$$\widetilde{T}_j^2 = \frac{[\mathbf{\Phi}\boldsymbol{\mu_y} - \boldsymbol{W}_* \boldsymbol{\mu_x}]_j^2}{n} \left\| \widehat{\boldsymbol{M}}_{\boldsymbol{xx}}^{-1/2} \left( \frac{\boldsymbol{x}^1}{\sqrt{n}} \mid \cdots \mid \frac{\boldsymbol{x}^n}{\sqrt{n}} \right) \mathbf{1}_n \right\|_2^2$$

$$= \frac{[\mathbf{\Phi}\boldsymbol{\mu_y} - \boldsymbol{W}_* \boldsymbol{\mu_x}]_j^2}{n} \cdot d$$

Therefore, the rescaled approximation error for the compressed version is upper-bounded by

$$\left\| \widehat{\mathbb{E}}[(\mathbf{\Phi}\boldsymbol{y} - \boldsymbol{W}_* \boldsymbol{x})\boldsymbol{x}^\top \widehat{\boldsymbol{M}}_{\boldsymbol{xx}}^{-1/2}] \right\|_F^2$$

$$\leq 2(\|\boldsymbol{W}_*\|_F^2 + \|\mathbf{\Phi}\|_F^2) \cdot \frac{d + 2\sqrt{d\log(m/\tau)} + 2\log(m/\tau)}{n} + 2\|\mathbf{\Phi}\boldsymbol{\mu_y} - \boldsymbol{W}_* \boldsymbol{\mu_x}\|_2^2 \cdot \frac{d}{n}$$

with probability at least $1 - \tau$. Similarly, it is easy to get that for any positive constant $\epsilon$, as

$$n \geq \widetilde{n}_2 := \max \left\{ 4(\|\boldsymbol{W}_*\|_F^2 + \|\mathbf{\Phi}\|_F^2) \cdot \frac{d + 2\sqrt{d\log(m/\tau)} + 2\log(m/\tau)}{\epsilon}, \ 4\|\mathbf{\Phi}\boldsymbol{\mu_y} - \boldsymbol{W}_* \boldsymbol{\mu_x}\|_2^2 \cdot \frac{d}{\epsilon} \right\},$$

we have $\left\| \widehat{\mathbb{E}}[(\mathbf{\Phi}\boldsymbol{y} - \boldsymbol{W}_* \boldsymbol{x})\boldsymbol{x}^\top \widehat{\boldsymbol{M}}_{\boldsymbol{xx}}^{-1/2}] \right\|_F^2 \leq \epsilon$ holds with probability at least $1 - \tau$.

**Step-3'.** Combining two upper bounds together, if $n \geq \max\{n_1, \widetilde{n}_2\}$, the following inequality for generalization error bound

$$\mathcal{L}^{\mathbf{\Phi}}(\widehat{\boldsymbol{W}}) - \mathcal{L}^{\mathbf{\Phi}}(\boldsymbol{W}_*) \leq \left\| \widehat{\mathbb{E}}[(\mathbf{\Phi}\boldsymbol{y} - \boldsymbol{W}_* \boldsymbol{x})\boldsymbol{x}^\top \widehat{\boldsymbol{M}}_{\boldsymbol{xx}}^{-1/2}] \right\|_F^2 \cdot \left\| \boldsymbol{M}_{\boldsymbol{xx}}^{1/2} \widehat{\boldsymbol{M}}_{\boldsymbol{xx}}^{-1} \boldsymbol{M}_{\boldsymbol{xx}}^{1/2} \right\|_{op} \leq 4\epsilon$$

holds with probability at least $1 - 3\tau$. $\qquad\square$

## A.4 Proof of Theorem 4

Recall the following theorem in the main text.

**Theorem 4** For any $\delta \in (0, 1)$ and any $\tau \in (0, 1/3)$, suppose the compressed matrix $\mathbf{\Phi}$ follows Assumption 1 with $m \geq O(\frac{s}{\delta^2} \log(\frac{K}{\tau}))$, and Assumption 2 holds. Given any learned regressor $\widehat{\boldsymbol{W}}$ from training problem (2), let $(\boldsymbol{x}, \boldsymbol{y})$ be a new sample drawn from the underlying distribution $\mathcal{D}$, we have the following inequality holds with probability at least $1 - \tau$:

$$\mathbb{E}_{\mathcal{D}}[\|\widehat{\boldsymbol{y}} - \boldsymbol{y}\|_2^2] \leq \frac{4}{1 - \delta} \cdot \mathbb{E}_{\mathcal{D}}[\|\mathbf{\Phi}\boldsymbol{y} - \widehat{\boldsymbol{W}}\boldsymbol{x}\|_2^2],$$

where $\widehat{\boldsymbol{y}}$ is the optimal solution from prediction problem (3) with input vector $\boldsymbol{x}$.

*Proof.* Due to the optimality of $\widehat{\boldsymbol{y}}$, we have

$$\left\| \mathbf{\Phi}\widehat{\boldsymbol{y}} - \widehat{\boldsymbol{W}}\boldsymbol{x} \right\|_2^2 \leq \left\| \mathbf{\Phi}\boldsymbol{y} - \widehat{\boldsymbol{W}}\boldsymbol{x} \right\|_2^2$$

$$\Leftrightarrow \left\| \mathbf{\Phi}\widehat{\boldsymbol{y}} - \mathbf{\Phi}\boldsymbol{y} + \mathbf{\Phi}\boldsymbol{y} - \widehat{\boldsymbol{W}}\boldsymbol{x} \right\|_2^2 \leq \left\| \mathbf{\Phi}\boldsymbol{y} - \widehat{\boldsymbol{W}}\boldsymbol{x} \right\|_2^2$$

$$\Leftrightarrow \| \mathbf{\Phi}\widehat{\boldsymbol{y}} - \mathbf{\Phi}\boldsymbol{y} \|_2^2 \leq 2\langle \mathbf{\Phi}\widehat{\boldsymbol{y}} - \mathbf{\Phi}\boldsymbol{y}, \mathbf{\Phi}\boldsymbol{y} - \widehat{\boldsymbol{W}}\boldsymbol{x} \rangle$$

$$\Rightarrow \| \mathbf{\Phi}\widehat{\boldsymbol{y}} - \mathbf{\Phi}\boldsymbol{y} \|_2^2 \leq 2 \| \mathbf{\Phi}\widehat{\boldsymbol{y}} - \mathbf{\Phi}\boldsymbol{y} \|_2 \left\| \mathbf{\Phi}\boldsymbol{y} - \widehat{\boldsymbol{W}}\boldsymbol{x} \right\|_2$$

$$\Leftrightarrow \| \mathbf{\Phi}\widehat{\boldsymbol{y}} - \mathbf{\Phi}\boldsymbol{y} \|_2 \leq 2 \left\| \mathbf{\Phi}\boldsymbol{y} - \widehat{\boldsymbol{W}}\boldsymbol{x} \right\|_2$$

$$\Leftrightarrow \| \mathbf{\Phi}\widehat{\boldsymbol{y}} - \mathbf{\Phi}\boldsymbol{y} \|_2^2 \leq 4 \left\| \mathbf{\Phi}\boldsymbol{y} - \widehat{\boldsymbol{W}}\boldsymbol{x} \right\|_2^2$$

$$\Rightarrow (1 - \delta)\|\widehat{\boldsymbol{y}} - \boldsymbol{y}\|_2^2 \leq 4 \left\| \mathbf{\Phi}\boldsymbol{y} - \widehat{\boldsymbol{W}}\boldsymbol{x} \right\|_2^2$$

where the final $\Rightarrow$ holds due to the $(3s, \delta)$-RIP property of the compressed matrix $\mathbf{\Phi}$ with probability at least $1 - \tau$. Taking expectations on both sides implies

$$(1 - \delta)\mathbb{E}_{\mathcal{D}}[\|\widehat{\boldsymbol{y}} - \boldsymbol{y}\|_2^2] \leq 4\mathbb{E}_{\mathcal{D}}[\|\mathbf{\Phi}\boldsymbol{y} - \widehat{\boldsymbol{W}}\boldsymbol{x}\|_2^2],$$

which completes the story. $\qquad\square$

## A.5 Technical Lemma

### A.5.1 Proof of Claim Proposed in Remark 2

*Proof.* Let us discuss the computational complexity for $\mathcal{F}$ to be $\mathbb{R}^K, \mathbb{R}_+^K, \{0,1\}^K$ separately. Given a fixed $\tilde{\boldsymbol{v}}$,

• If $\mathcal{F} = \mathbb{R}^K$, the projection method $\arg\min_{\boldsymbol{v} \in \mathcal{V}_s^K \cap \mathcal{F}} \|\boldsymbol{v} - \tilde{\boldsymbol{v}}\|_2^2$ can be reformulate using the following mixed-integer programming (MIP),

$$(\boldsymbol{v}_*, \boldsymbol{z}_*) := \arg\min_{\boldsymbol{v},\boldsymbol{z}} \quad \sum_{p=1}^K z_p(\boldsymbol{v}_p - \tilde{\boldsymbol{v}}_p)^2 \atop \text{s.t.} \quad \sum_{p=1}^K z_p \leq s \quad,$$

with $\boldsymbol{v}_*$ the output of the projection method. Sorting the absolute values $\{|\tilde{\boldsymbol{v}}_p|\}_{p=1}^K$ in decreasing order such that

$$|\tilde{\boldsymbol{v}}_{(1)}| \geq \cdots \geq |\tilde{\boldsymbol{v}}_{(K)}|,$$

the output $\boldsymbol{v}_*$ of the proposed projection is

$$[\boldsymbol{v}_*]_j = \begin{cases} \tilde{\boldsymbol{v}}_j & \text{if } j \in \{(1), \ldots, (s)\} \\ 0 & \text{o.w.} \end{cases} \quad,$$

with computational complexity $O(K\min\{s, \log K\})$.

• If $\mathcal{F} = \mathbb{R}_+^K$, the projection method $\arg\min_{\boldsymbol{v} \in \mathcal{V}_s^K \cap \mathcal{F}} \|\boldsymbol{v} - \tilde{\boldsymbol{v}}\|_2^2$ can be reformulate using the following mixed-integer programming (MIP),

$$(\boldsymbol{v}_*, \boldsymbol{z}_*) := \arg\min_{\boldsymbol{v},\boldsymbol{z}} \quad \sum_{p=1}^K z_p(\boldsymbol{v}_p - \tilde{\boldsymbol{v}}_p)^2 \atop \begin{array}{l} \text{s.t.} \quad \sum_{p=1}^K z_p \leq s \\ \quad \boldsymbol{v}_p \geq 0 \; \forall\, p \in [K] \end{array} \quad.$$

Sorting $\{\tilde{\boldsymbol{v}}_p\}_{p=1}^K$ in decreasing order such that

$$\tilde{\boldsymbol{v}}_{(1)} \geq \cdots \geq \tilde{\boldsymbol{v}}_{(K)},$$

the output $\boldsymbol{v}_*$ of the proposed projection is

$$[\boldsymbol{v}_*]_j = \begin{cases} \tilde{\boldsymbol{v}}_j \cdot \mathbb{I}(\tilde{\boldsymbol{v}}_j > 0) & \text{if } j \in \{(1), \ldots, (s)\} \\ 0 & \text{o.w.} \end{cases} \quad,$$

with computation complexity $O(K\min\{s, \log K\})$.

• If $\mathcal{F} = \{0,1\}^K$, the projection method $\min_{\boldsymbol{v} \in \mathcal{V}_s^K \cap \mathcal{F}} \|\boldsymbol{v} - \tilde{\boldsymbol{v}}\|_2^2$ presented in step-4 of Algorithm 1 can be represented as

$$\begin{array}{l} \min_{\boldsymbol{z}} \quad \sum_{p=1}^K (1 - z_p)\tilde{\boldsymbol{v}}_p^2 + z_p(\tilde{\boldsymbol{v}}_p - 1)^2 \\ \text{s.t.} \quad \sum_{p=1}^K z_p \leq s \\ \quad z_p \in \{0,1\} \; \forall\, p \in [K] \end{array} \quad = \quad \begin{array}{l} \min_{\boldsymbol{z}} \quad \sum_{p=1}^K \tilde{\boldsymbol{v}}_p^2 - z_p(2\tilde{\boldsymbol{v}}_p - 1) \\ \text{s.t.} \quad \sum_{p=1}^K z_p \leq s \\ \quad z_p \in \{0,1\} \; \forall\, p \in [K] \end{array} \quad.$$

Sort $\{2\tilde{\boldsymbol{v}}_p - 1\}_{p=1}^K$ in decreasing order such that

$$2\tilde{\boldsymbol{v}}_{(1)} - 1 \geq 2\tilde{\boldsymbol{v}}_{(2)} - 1 \geq \cdots \geq 2\tilde{\boldsymbol{v}}_{(K)} - 1,$$

then, the optimal $\boldsymbol{z}^*$ can be set by

$$z_p^* = \begin{cases} \mathbb{I}(2\boldsymbol{v}_p - 1 > 0) & \text{if } p \in \{(1), \ldots, (s)\} \\ 0 & \text{o.w.} \end{cases} \quad.$$

For computational complexity, computing the sequence $\{2\boldsymbol{v}_p - 1\}_{p=1}^K$ needs $O(K)$, picking the top-$s$ elements of the above sequence requires $O(K\min\{s, \log K\})$, setting the optimal solution $\boldsymbol{z}^*$ needs $O(s)$, and thus the total computational complexity is $O(K) + O(K\min\{s, \log K\}) + O(s) = O(K\min\{s, \log K\})$. □

### A.5.2 Lemma for the Proof of Theorem 1

**Lemma 1.** *(Unitary invariant). Let $\boldsymbol{\Phi} \in \mathbb{R}^{m \times d}$ be a randomized compressed matrix as described in Assumption 1, and $\boldsymbol{U} \in \mathbb{R}^{d \times d}$ be a real unitary matrix. Then we have $\tilde{\boldsymbol{\Phi}} = \boldsymbol{\Phi}\boldsymbol{U}$ is $(1, \delta)$-RIP with probability at least $1 - \tau$.*

*Proof.* Note that $(i, j)$-th component of $\tilde{\boldsymbol{\Phi}}$ can be represented as $\tilde{\boldsymbol{\Phi}}_{i,j} = \boldsymbol{\Phi}_{i,:}\boldsymbol{U}_{:,j} = \sum_{\ell=1}^{d} \boldsymbol{\Phi}_{i,\ell}\boldsymbol{U}_{\ell,j}$. Since every component $\boldsymbol{\Phi}_{i,j}$ in $\boldsymbol{\Phi}$ is i.i.d. drawn from $\mathcal{N}(0, 1/m)$, we have

$$\tilde{\boldsymbol{\Phi}}_{i,j} = \sum_{\ell=1}^{d} \boldsymbol{\Phi}_{i,\ell}\boldsymbol{U}_{\ell,j} \sim \mathcal{N}\left(0, \sum_{\ell=1}^{d} \frac{1}{m}\boldsymbol{U}_{\ell,j}^2\right) = \mathcal{N}(0, 1/m).$$

Now, we need to show that any two distinct components $\tilde{\boldsymbol{\Phi}}_{i_1,j_1}$ and $\tilde{\boldsymbol{\Phi}}_{i_2,j_2}$ in $\tilde{\boldsymbol{\Phi}}$ are independent. It is easy to observe that $\tilde{\boldsymbol{\Phi}}_{i_1,j_1}$ and $\tilde{\boldsymbol{\Phi}}_{i_2,j_2}$ are independent when $i_1 \neq i_2$ since $\boldsymbol{\Phi}_{i_1,:}$ and $\boldsymbol{\Phi}_{i_2,:}$ are independent. If $i_1 = i_2 = i$, then the following random vector satisfies

$$\begin{pmatrix} \tilde{\boldsymbol{\Phi}}_{i,j_1} \\ \tilde{\boldsymbol{\Phi}}_{i,j_1} \end{pmatrix} = \begin{pmatrix} \boldsymbol{\Phi}_{i,:}\boldsymbol{U}_{:,j_1} \\ \boldsymbol{\Phi}_{i,:}\boldsymbol{U}_{:,j_2} \end{pmatrix} \sim \mathcal{N}\left(\begin{pmatrix} 0 \\ 0 \end{pmatrix}, \begin{pmatrix} 1/m & 0 \\ 0 & 1/m \end{pmatrix}\right).$$

That is to say, $\tilde{\boldsymbol{\Phi}}_{i_1,j_1}$ and $\tilde{\boldsymbol{\Phi}}_{i_2,j_2}$ are *jointly Gaussian distributed and uncorrelated*, which shows that $\tilde{\boldsymbol{\Phi}}_{i_1,j_1}$ and $\tilde{\boldsymbol{\Phi}}_{i_2,j_2}$ are independent. Combining the above together, we have $\tilde{\boldsymbol{\Phi}}$ is a randomized matrix with component i.i.d. from $\mathcal{N}(0, 1/m)$. Based on the existing result ([7], Theorem 1.5), when $m \geq C_1 \cdot \delta^{-2}[\ln(eK) + \ln(2/\tau)]$ for any $\delta > 0$ and $\tau \in (0, 1)$, we have $\tilde{\boldsymbol{\Phi}}$ ensures $(1, \delta)$-RIP with probability at least $1 - \tau$. $\qquad\square$

### A.5.3 Lemma for the Proof of Theorem 2

**Lemma 2.** *For any integer parameter $s(\leq d)$ and positive parameter $\delta \in (0, 1)$, let $\boldsymbol{\Phi} \in \mathbb{R}^{m \times d}$ be a $(s, \delta)$-RIP matrix. For $\boldsymbol{u}_1, \boldsymbol{u}_2$, if $\boldsymbol{u}_1, \boldsymbol{u}_2, \boldsymbol{u}_1 + \boldsymbol{u}_2, \boldsymbol{u}_1 - \boldsymbol{u}_2$ are all $s$-sparse, then the following inequality holds*

$$-2\delta(\|\boldsymbol{u}_1\|_2^2 + \|\boldsymbol{u}_2\|_2^2) + 4\langle\boldsymbol{u}_1, \boldsymbol{u}_2\rangle \leq 4\langle\boldsymbol{\Phi}\boldsymbol{u}_1, \boldsymbol{\Phi}\boldsymbol{u}_2\rangle \leq 2\delta(\|\boldsymbol{u}_1\|_2^2 + \|\boldsymbol{u}_2\|_2^2) + 4\langle\boldsymbol{u}_1, \boldsymbol{u}_2\rangle.$$

*Proof.* Since $\boldsymbol{u}_1, \boldsymbol{u}_2, \boldsymbol{u}_1 + \boldsymbol{u}_2, \boldsymbol{u}_1 - \boldsymbol{u}_2$ are $s$-sparse, we have

$$(1 - \delta)(\|\boldsymbol{u}_1 + \boldsymbol{u}_2\|_2^2) \leq \langle\boldsymbol{\Phi}(\boldsymbol{u}_1 + \boldsymbol{u}_2), \boldsymbol{\Phi}(\boldsymbol{u}_1 + \boldsymbol{u}_2)\rangle \leq (1 + \delta)(\|\boldsymbol{u}_1 + \boldsymbol{u}_2\|_2^2) \qquad (8)$$

$$(1 - \delta)(\|\boldsymbol{u}_1 - \boldsymbol{u}_2\|_2^2) \leq \langle\boldsymbol{\Phi}(\boldsymbol{u}_1 - \boldsymbol{u}_2), \boldsymbol{\Phi}(\boldsymbol{u}_1 - \boldsymbol{u}_2)\rangle \leq (1 + \delta)(\|\boldsymbol{u}_1 - \boldsymbol{u}_2\|_2^2) \qquad (9)$$

Subtracting (9) from (8) gives

$$-2\delta(\|\boldsymbol{u}_1\|_2^2 + \|\boldsymbol{u}_2\|_2^2) + 4\langle\boldsymbol{u}_1, \boldsymbol{u}_2\rangle \leq 4\langle\boldsymbol{\Phi}\boldsymbol{u}_1, \boldsymbol{\Phi}\boldsymbol{u}_2\rangle \leq 2\delta(\|\boldsymbol{u}_1\|_2^2 + \|\boldsymbol{u}_2\|_2^2) + 4\langle\boldsymbol{u}_1, \boldsymbol{u}_2\rangle,$$

which completes the proof. $\qquad\square$

### A.5.4 Lemma for the Proof of Theorem 3

**Lemma 3.** *Let $\boldsymbol{\xi}^1, \ldots, \boldsymbol{\xi}^n$ be $n$ i.i.d. $\sigma^2$-subGaussian random vectors with a zero mean and a covariance matrix $\boldsymbol{\Sigma}$. Then, there exists a positive constant $C$ such that for all $\tau \in (0, 1)$,*

$$\mathbb{P}\left(\left\|\sum_{i=1}^{n} \frac{\boldsymbol{\xi}^i(\boldsymbol{\xi}^i)^\top}{n} - \boldsymbol{\Sigma}\right\|_{op} \leq C\sigma^2 \max\left\{\sqrt{\frac{d + \log(2/\tau)}{n}}, \frac{d + \log(2/\tau)}{n}\right\}\right) \geq 1 - \tau.$$

*Proof.* Lemma 3 is a direct corollary from [Theorem 6.5, [41]]. It is easy to observe that the proposed Lemma 3 holds by setting the parameter $\delta$ listed in [Theorem 6.5, [41]] as $\min\{1, c\sqrt{\ln(2/\tau)/n}\}$ with $c$ some positive constant. $\qquad\square$

**Lemma 4.** *Let $\boldsymbol{\xi}^1, \ldots, \boldsymbol{\xi}^n$ be $n$ i.i.d. $\sigma^2$-subGaussian random vectors with a zero mean and a covariance matrix $\boldsymbol{\Sigma}$. Then for any $\tau \in (0, 1)$, we have*

$$\mathbb{P}\left(\left\|\sum_{i=1}^{n} \frac{\boldsymbol{\xi}^i}{n}\right\|_2 \leq \frac{4\sigma\sqrt{d} + 2\sigma\sqrt{\log(1/\tau)}}{\sqrt{n}}\right) \geq 1 - \tau.$$

*Proof.* We show this lemma by discretizing the unit $\ell_2$-norm ball $\mathbb{B}_2(\mathbf{0}; 1)$. Let $\mathcal{N}_{1/2}$ be a $\frac{1}{2}$-minimum cover of $\mathbb{B}_2(\mathbf{0}; 1)$ with its cardinality $|\mathcal{N}_{1/2}| \leq 5^d$. Since for any vector $\boldsymbol{\xi} \in \mathbb{R}^d$, we always have

$$\|\boldsymbol{\xi}\|_2 = \max_{\|\boldsymbol{v}\|_2 \leq 1} \langle \boldsymbol{v}, \boldsymbol{\xi} \rangle \leq \max_{\boldsymbol{v}' \in \mathcal{N}_{1/2}} \langle \boldsymbol{v}', \boldsymbol{\xi} \rangle + \max_{\|\boldsymbol{v}''\|_2 \leq 1/2} \langle \boldsymbol{v}'', \boldsymbol{\xi} \rangle = \max_{\boldsymbol{v}' \in \mathcal{N}_{1/2}} \langle \boldsymbol{v}', \boldsymbol{\xi} \rangle + \frac{1}{2} \max_{\|\boldsymbol{v}''\|_2 \leq 1} \langle \boldsymbol{v}'', \boldsymbol{\xi} \rangle,$$

then $\|\boldsymbol{\xi}\|_2 \leq 2 \max_{\boldsymbol{v}' \in \mathcal{N}_{1/2}} \langle \boldsymbol{v}', \boldsymbol{\xi} \rangle$. Therefore, for any $\sigma^2$-subGaussian random vector,

$$\mathbb{P}(\|\boldsymbol{\xi}\|_2 \geq t) \geq \mathbb{P}\left( \max_{\boldsymbol{v}' \in \mathcal{N}_{1/2}} \langle \boldsymbol{v}', \boldsymbol{\xi} \rangle \geq t/2 \right) \leq |\mathcal{N}_{1/2}| \cdot \exp\left( -\frac{t^2}{8\sigma^2} \right) \leq 5^d \cdot \exp\left( -\frac{t^2}{8\sigma^2} \right),$$

which implies that $\|\boldsymbol{\xi}\|_2 \leq 4\sigma\sqrt{d} + 2\sigma\sqrt{\log(1/\tau)}$ with probability at least $1 - \tau$. Now, since $\bar{\boldsymbol{\xi}} = \sum_{i=1}^n \frac{\boldsymbol{\xi}^i}{n}$ is a $\sigma^2/n$-subGaussian random vector, inserting this variance proxy into the above inequality gives the desired result. $\qquad\square$

**Lemma 5.** *Let $\eta(\boldsymbol{x})$ be a zero-mean, $\sigma_\eta^2$-subGaussian random variable. Let $\boldsymbol{x}^1, \ldots, \boldsymbol{x}^n$ be $n$ i.i.d. $\sigma^2$-subGaussian random vectors (may not zero-mean) as described in Assumption 2. Conditioned on $\widehat{\boldsymbol{M}}_{\boldsymbol{xx}} = \frac{1}{n} \sum_{i=1}^n \boldsymbol{x}^i (\boldsymbol{x}^i)^\top \succ \mathbf{0}_{d \times d}$, for any $\tau \in (0,1)$, we have*

$$\mathbb{P}\left( \left\| \widehat{\mathbb{E}}[\eta(\boldsymbol{x})\boldsymbol{x}^\top \widehat{\boldsymbol{M}}_{\boldsymbol{xx}}^{-1/2}] \right\|_2^2 \leq \frac{\sigma_\eta^2 (d + 2\sqrt{d\log(1/\tau)} + 2\log(1/\tau))}{n} \right) \geq 1 - \tau.$$

*Proof.* The proof of Lemma 5 can be found in [Lemma 5, [18]]. $\qquad\square$

### A.5.5 Discussion after Theorem 4

Since $\|\boldsymbol{v}^{(T)} - \boldsymbol{y}\|_2 \leq \|\boldsymbol{v}^{(T)} - \widehat{\boldsymbol{y}}\|_2 + \|\widehat{\boldsymbol{y}} - \boldsymbol{y}\|_2$, combined with Theorem 2, we have

$$\begin{aligned}
\mathbb{E}_{\mathcal{D}}[\|\boldsymbol{v}^{(T)} - \boldsymbol{y}\|_2^2] &\leq 2\mathbb{E}_{\mathcal{D}}[\|\boldsymbol{v}^{(T)} - \widehat{\boldsymbol{y}}\|_2^2] + 2\mathbb{E}_{\mathcal{D}}[\|\widehat{\boldsymbol{y}} - \boldsymbol{y}\|_2^2] \\
&\leq O(\mathbb{E}_{\mathcal{D}}[\|\boldsymbol{\Phi}\widehat{\boldsymbol{y}} - \widehat{\boldsymbol{W}}\boldsymbol{x}\|_2]) + \frac{8}{1-\delta} \cdot \mathbb{E}_{\mathcal{D}}[\|\boldsymbol{\Phi}\boldsymbol{y} - \widehat{\boldsymbol{W}}\boldsymbol{x}\|_2^2] \\
&\leq O(\mathbb{E}_{\mathcal{D}}[\|\boldsymbol{\Phi}\boldsymbol{y} - \widehat{\boldsymbol{W}}\boldsymbol{x}\|_2]).
\end{aligned}$$

## A.6 Additional Numerical Experiments on Synthetic Data

### A.6.1 Implemented Prediction Method

The implemented prediction method is presented as follows, see Algorithm 2. Comparing with Algorithm 1 proposed in the main content, it adds an additional stopping criteria

$$\frac{\|\boldsymbol{v}^{(t)} - \boldsymbol{v}^{(t-2)}\|_2}{0.01 + \|\boldsymbol{v}^{(t)}\|_2} < 10^{-6}$$

to ensure an earlier stop than Algorithm 1.

Note, in later numerical experiments, we use the terminology *'early stopping'* to denote that the iteration generated by the prediction algorithm satisfies the above additional stopping criteria within the maximum iteration number, i.e. $T = 60$ (as listed in Section 4).

### A.6.2 Discussions on Baselines

*Baselines.* We compare our proposed prediction method with the following baselines.

**Orthogonal Matching Pursuit.** Orthogonal Matching Pursuit(OMP) is a greedy prediction algorithm. It iteratively chooses the most relevant output and then performs least-squares and updates the residuals. The built-in function 'OrthogonalMatchingPursuit' from Python package 'Sklearn.Linear_model' is used in the experiment.

**Correlation Decoding.** Correlation decoding is a standard decoding algorithm. It computes the multiplication of the transpose of compression matrix $\boldsymbol{\Phi}$ and the learned regressor $\widehat{\boldsymbol{W}}$. For any test point $\boldsymbol{x}$, the algorithm predicts the top $s$ labels in $\boldsymbol{\Phi}\widehat{\boldsymbol{W}}\boldsymbol{x}$ ordered by magnitude.

---

**Algorithm 2** Implemented Projected Gradient Descent (for Second Stage)

---

**Input:** Regressor $\widehat{\boldsymbol{W}}$, input sample $\boldsymbol{x}$, stepsize $\eta$, total iterations $T$

1: **Initialize** point $\boldsymbol{v}^{(0)} \in \mathcal{V}_s^K \cap \mathcal{F}$ and $t = 0$.
2: **while** $t < T$ **and** $\|\boldsymbol{v}^{(t)} - \boldsymbol{v}^{(t-2)}\|_2 / (0.01 + \|\boldsymbol{v}^{(t)}\|_2) > 10^{-3}$: **do**
3:     Update $\widetilde{\boldsymbol{v}}^{(t+1)} = \boldsymbol{v}^{(t)} - \eta \cdot \boldsymbol{\Phi}^\top (\boldsymbol{\Phi}\boldsymbol{v}^{(t)} - \widehat{\boldsymbol{W}}\boldsymbol{x})$.
4:     Project $\boldsymbol{v}^{(t+1)} = \Pi(\widetilde{\boldsymbol{v}}^{(t+1)}) := \arg\min_{\boldsymbol{v} \in \mathcal{V}_s^K \cap \mathcal{F}} \|\boldsymbol{v} - \widetilde{\boldsymbol{v}}^{(t+1)}\|_2^2$.
5:     Update $t := t + 1$.
6: **end while**

**Output:** $\boldsymbol{v}^{(T)}$.

---

**Elastic Net.** The elastic net is a regression method that combines both the $\ell_1$-norm penalty and the $\ell_2$-norm penalty to guarantee the sparsity and the stability of the prediction. The parameters for the $\ell_1$-norm penalty and $\ell_1$-norm penalty in Elastic Net are set to be 0.1 through the experiments. The built-in function 'ElasticNet' from Python package 'Sklearn.Linear_model' is used in the experiment.

**Fast Iterative Shrinkage-Thresholding Algorithm.** The fast iterative shrinkage-thresholding algorithm is an advanced optimization method designed to efficiently solve certain classes of unconstrained convex optimization problems. It utilizes momentum-like strategies to speed up convergence rates, which is particularly effective for minimizing functions that may be non-smooth.

### A.6.3 Procedures on Generating Mean & Covariance

In this paper, we give exact procedures on selecting $\boldsymbol{\mu_x}$ and $\boldsymbol{\Sigma_{xx}}$ as mentioned in Section 4.

• The mean vector $\boldsymbol{\mu_x}$ is selected as follows. We first generate a $d$-dimensional vector $\boldsymbol{\mu}$ from the standard $d$-dimensional Gaussian distribution $\mathcal{N}(\boldsymbol{0}_d, \boldsymbol{I}_d)$. Then we set

$$[\boldsymbol{\mu_x}]_j = |\boldsymbol{\mu}|_j \ \text{ for all } \ j \in [d]$$

by taking absolute values over all components.

• The covariance matrix $\boldsymbol{\Sigma_{xx}}$ is selected as follows. We first generate a $d$-by-$d$ matrix $\boldsymbol{A}$, where every component of $\boldsymbol{A}$ is i.i.d. generated from the standard Gaussian distribution $\mathcal{N}(0, 1)$. Then the covariance matrix $\boldsymbol{\Sigma_{xx}}$ is set to be

$$\boldsymbol{\Sigma_{xx}} := \frac{1}{d}\boldsymbol{A}^\top \boldsymbol{A} + \frac{1}{2}\boldsymbol{I}_d.$$

### A.7 Additional Numerical Experiments on Real Data

In this subsection, we do experiments on real data and compare the performance of the proposed prediction method (see Algorithm 2) with four baselines, i.e., Orthogonal Matching Pursuit (OMP, [46]), Correlation Decoding (CD,[20]), Elastic Net (EN, [47]), and Fast Iterative Shrinkage-Thresholding Algorithm (FISTA,[2]) see Appendix A.6.2 for detailed explanations.

**Real data.** We select two benchmark datasets in multi-label classification, Wiki10-31K and EURLex-4K[5] due to their sparsity property. Table 1 shows the details for the datasets.

| Dataset | Input Dim $d$ | Output Dim $K$ | Training set $n$ | $\overline{d}$ | $\overline{K}$ | Test set $n$ | $\overline{d}$ | $\overline{K}$ |
|---|---|---|---|---|---|---|---|---|
| **EURLex-4K** | 5,000 | 3,993 | 17,413 | 236.69 | 5.30 | 1,935 | 240.96 | 5.32 |
| **Wiki10-31K** | 101,938 | 30,938 | 14,146 | 673.45 | 18.64 | 6,616 | 659.65 | 19.03 |

Table 1: Statistics and details for training and test sets, where $\overline{d}, \overline{K}$ denote their averaged non-zero components for input and output, respectively.

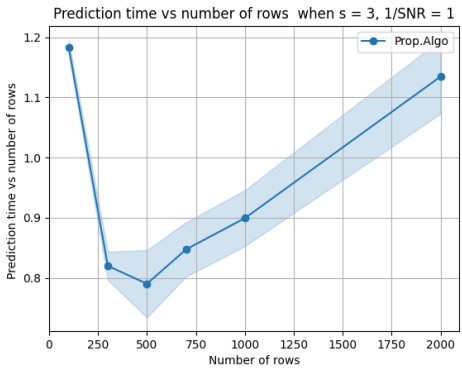

Figure 2: The figure reports the prediction running time (measured in seconds) on synthetic data with early stopping by the proposed algorithm under different compressed output dimensions. As we can observe, the running time first decreases dramatically, then increases almost linearly with respect to $m$. Such a phenomenon has occurred since the max number of iterations is 60 in the implemented prediction method with early stopping, which is relatively large; As $m$ increases but is still less than 500, the actual number of iterations drops dramatically due to early stopping criteria; After passes 500, the actual number of iterations stays around 10, and then the running time grows linearly as dimension increases.

**Parameter setting.** In prediction stage, we choose $s \in \{1,3\}$ for EURLex-4K and $s \in \{3,5\}$ for Wiki10-31K. We choose the number of rows $m \in \{100, 200, 300, 400, 500, 700, 1000, 2000\}$ on both EURLex-4K and Wiki10-31. Ten independent trials of compressed matrices $\mathbf{\Phi}^{(1)}, \ldots, \mathbf{\Phi}^{(10)}$ are implemented for each tuple of parameters $(s, m)$ on both datasets.

**Empirical running time.** Here, we report the running time of the proposed algorithm and baselines on both synthetic and real datasets, see Table 2.

| Dataset | Prop.Algo. | OMP | CD | Elastic Net | FISTA |
|---|---|---|---|---|---|
| **Synthetic Data** | ≈1 second | 200-400 seconds | <1 second | 700-900 seconds | <3 seconds |
| **EURLex-4K** | <1 second | 20-80 seconds | <1 second | - | ≈ 1 second |
| **Wiki10-31K** | <5 seconds | 500-700 seconds | <5 seconds | - | 5-10 seconds |

Table 2: Time Complexity Comparison for each prediction

**Numerical Results & Discussions.** Figure 2 further illustrates that the computational complexity increases linearly with respect to the growth of compressed output dimension $m$ on synthetic data, when $m$ is greater than $500$ to ensure the convergence of the prediction algorithm (see Remark 2).

For real data, Figure 3 and Figure 4 present the results of their accuracy performances. In particular, the accuracy grows relatively stable with respect to $m$ when the compression matrix satisfies the RIP-property with high probability. Besides, based on the results presented in Figure 3, Figure 4, and Table 2, we observe that the proposed algorithm slightly outperforms the baselines on precision as $s$ increases while enjoys a significant better computational efficiency, especially on large instances, which demonstrate the stability of the proposed algorithm.

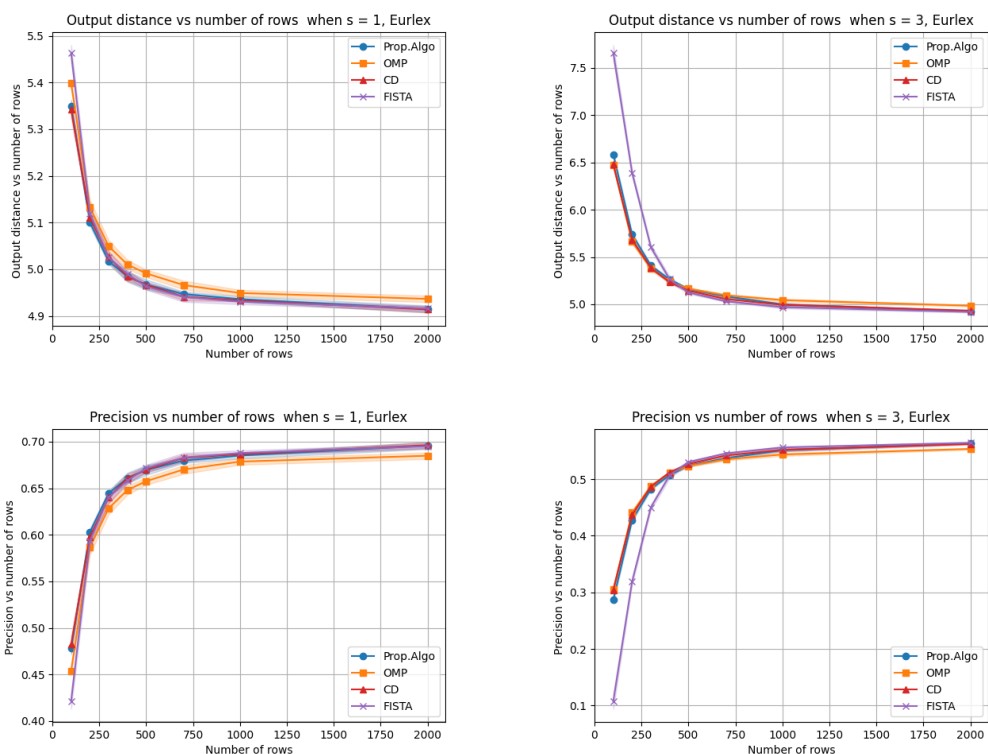

Figure 3: This figure reports the numerical results on real data – EURLex-4K. Each dot in the figure represents 10 *independent trials* (i.e., experiments) of compressed matrices $\mathbf{\Phi}^{(1)}, \ldots, \mathbf{\Phi}^{(10)}$ on a given tuple of parameters $(s, m)$. The curves in each panel correspond to the averaged values for the proposed Algorithm and baselines over 10 trials; the shaded parts represent the empirical standard deviations over 10 trials. In the first row, we plot the output distance versus the number of rows. In the second row, we plot the precision versus the number of rows, and we cannot observe significant differences between these prediction methods.

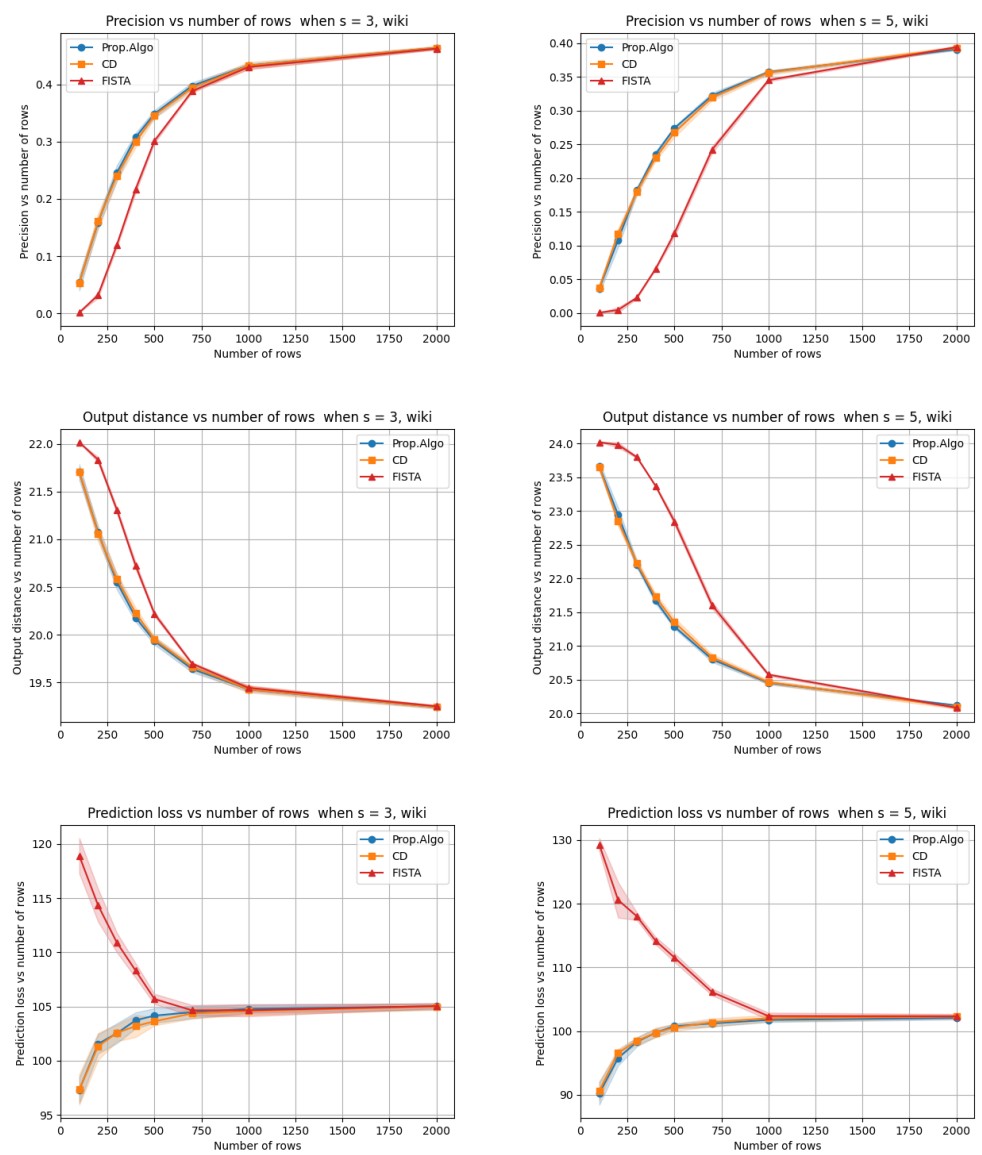

Figure 4: This figure reports the numerical results on real data – Wiki10-31K. Similar to the plot reporting on EURLex-4K above, each dot in the figure represents 10 *independent trials* (i.e., experiments) of compressed matrices $\mathbf{\Phi}^{(1)}, \dots, \mathbf{\Phi}^{(10)}$ on a given tuple of parameters $(s, m)$. The curves in each panel correspond to the averaged values for the proposed algorithm and baselines over 10 trials; the shaded parts represent the empirical standard deviations over 10 trials. Similarly, in the first row, the precision of the proposed algorithm outperforms the FISTA especially when $s$ is small. In the second & third rows for output difference and prediction loss, there are only slight improvement on the proposed algorithm than CD of output difference.

