# OpenReview forum: "Solving Sparse \& High-Dimensional-Output Regression via Compression"
_NeurIPS.cc/2024/Conference — NeurIPS 2024 poster_

### Official Review · Reviewer_PtcT · 2024-07-09

**Soundness:** 3
**Presentation:** 3
**Contribution:** 3
**Rating:** 7
**Confidence:** 4

**Summary:**

The paper proposes the Sparse & High-dimensional-Output REgression (SHORE) model to address challenges in Multi-Output Regression (MOR). SHORE introduces a two-stage framework to handle high-dimensional outputs efficiently. Theoretical analysis shows that the framework maintains training and prediction loss while being computationally scalable. Empirical results validate these findings, demonstrating the framework’s efficiency and accuracy in handling SHORE for modern MOR applications.

**Strengths:**

- Overall, this paper is well-written and well-organized.

- The two-stage algorithm proposed in the paper is easy to grasp and implement. Notably, by introducing a random projection matrix, it reduces the dimension of $Y$ and largely relaxes the RIP-type condition in sparsity-constrained optimization.

- The theoretical analysis is solid and includes a high-level interpretation of the proof. The imposed assumptions are mild and likely to (approximately) hold in practice.

**Weaknesses:**

- The prediction stage is significantly different from the classical manner (i.e., $Z X$). It involves solving sparsity-constrained optimization (SCO) problems. It would be good to highlight this difference. Additionally, solving SCO is more computationally intensive than the classical method, so more discussion (in both theory and experiments) is necessary.

- Algorithm 1: Algorithm 1: Steps 3-4 resemble iterative hard thresholding [1, 2]. It would be better to discuss this in the appropriate place.

- Lines 217-218: The setting for $y$ seems to imply that only $s$ columns have non-zero values while the remaining columns always take zero values. Under this assumption, why not filter out some columns according to the magnitude of the columns of $y$?

- It would be better to explicitly describe how OMP is compared. Is it used for solving (3) or (1)?

- Line 310-311: Elastic net may have better performance in a low signal regime [3].

- Considering these assumptions are mild in practice, why not test the algorithm on real-world datasets and present the results in the main text?

- Lines 437, 467, 542: Incomplete sentences. Perhaps they can be changed to "Recall the following theorem in the main text."

**Questions:**

- Theorem 4: $\hat{y}$ in the theorem is the optimal solution of (3). How about the algorithmic solution?

- The selection $\eta$ is much easier than in [2]. However, it would be good to provide some default values for practical usage. Furthermore, is it possible to select $\eta$ in a more data-driven way, for example, by leveraging the value of the loss function like in [4]?

**Limitations:**

The authors have discussed the limitations in this paper.

### Reference

- [1] Blumensath, Thomas, and Mike E. Davies. "Iterative hard thresholding for compressed sensing." Applied and computational harmonic analysis 27.3 (2009): 265-274.

- [2] Jain, Prateek, Ambuj Tewari, and Purushottam Kar. "On iterative hard thresholding methods for high-dimensional m-estimation." Advances in neural information processing systems 27 (2014).

- [3] Hastie, Trevor, Robert Tibshirani, and Ryan Tibshirani. "Best subset, forward stepwise or lasso? Analysis and recommendations based on extensive comparisons." Statistical Science 35.4 (2020): 579-592.

- [4] Wang, Zezhi, et al. "Sparsity-Constraint Optimization via Splicing Iteration." arXiv preprint arXiv:2406.12017 (2024).

---

> ### Author Rebuttal · Authors · 2024-08-06
>
> ## Response
>
> We sincerely thank the reviewer for their positive and encouraging feedback! Table 1 mentioned below can be found in the uploaded pdf file in 'global' author rebuttal.
>
> **[Response to Comparison with SCO]**
>
> Thank you for highlighting this aspect! If we understand correctly, the classical manner you mentioned denotes predicting SHORE *without compression*. Please refer to the 'global' author rebuttal part ''Proposed prediction stage v.s. General SCO'' and ''Proposed prediction stage v.s. Classical manner'' for detailed discussions. We will add these discussions in the introduction and problem setup sections.
>
> **[Response to Comparison with iterative hard thresholding]**
>
> Thank you for your suggestion. We will add the following discussion to Section 3.1 after introducing the proposed two-stage framework SHORE. Note that both the iterative hard thresholding methods (Blumensath et al. 2009, Jain et al. 2014) and the proposed Algorithm 1 share the same insight from the vanilla projected gradient method for nonconvex constrained minimization.
> Compared with existing iterative hard thresholding, Step 4 (projection step) projects the resulting vector to a more general feasible set $\mathcal{V}_s^K \cap \mathcal{F}$ with additional constraints other than sparsity, which might require non-trivial modifications in designing projection oracles and convex analysis. In particular, we discuss the different projection methods under different $\mathcal{F}$ through the perspective of optimization (see Appendix A.5.1 on page 19). We also discuss the bound for stepsize $\eta$ to guarantee the algorithm's convergence.
>
> **[Response to nonzero of y]**
>
> Thank you for raising this ambiguity! We only assume that each column of $\boldsymbol{Y}$, i.e., $\boldsymbol{y}^{i}$ for $i = 1, \ldots, n$, has $s$ non-zero components. However, the support sets of two distinct samples, e.g., $\boldsymbol{y}^{i}$ and $\boldsymbol{y}^{i'}$ with $i \neq i'$, might be different, which does not imply the row-sparsity for matrix $\boldsymbol{Y}$ (i.e., only $s$ rows/labels are non-zero).
>
> **[Response to Comparison with OMP]**
>
> Thank you for pointing this out! We will add an explicit description of both the OMP algorithm and how we compare the OMP algorithm with our algorithm in the camera-ready version.
> Here, we would like to point out that OMP is an algorithm to solve the sparse regression problem, which has some differences from the proposed problem in the prediction stage. See the global rebuttal part for details.
>
> **[Response to Comparison with Elastic Net]**
>
> Thank you for your insightful observation! We conducted several small experiments and found that Elastic Net works better in a low SNR regime (Refer to Table 1). We will add relevant comparisons in the revision.
>
> **[Response to testing on real-world dataset]**
>
> Thank you for identifying this point. Because of resource limitations, we only conducted several experiments on relatively small real datasets where  $\text{K} \approx 31,000$ , which are shown in Appendix 7. Currently, we are preparing more computational resources to conduct experiments on larger real-world datasets.
>
> **[Response to typos]**
>
> Thank you for pointing this out! We will correct them in the revision and do more proofreading.
>
> **[Response to $\hat{y}$ and algorithmic solution ]**
>
> Thank you for asking this! We actually discussed the algorithmic solution, denoted by $\boldsymbol{v}^{(T)}$​, a bit below Theorem 4, see lines 273 - 275. We will highlight the results in a remark in the revision.
>
> **[Response to selection of $\eta $]**
>
> In our numerical experiments, the default setting for $\eta$ is $\eta = 0.9$, which satisfies the convergence condition proposed in Theorem 2 and Theorem 4 for all $\delta \in (0, \frac{4}{9})$.
>
> For a data-driven setting on $\eta$, we realize that (Wang et al. 2024) was recently accessible on arXiv after the NeurIPS 2024's submission deadline, which introduces a novel iterative algorithm with a tuning-free property and linear convergence guarantee. We believe that this is a good related future direction and will be discussed in Section 5.
>
>
>
> ## Reference
>
> **[Blumensath et al. 2009]**
> Thomas Blumensath and Mike E Davies. Iterative hard thresholding for compressed sensing.
> Applied and computational harmonic analysis, 27(3):265–274, 2009.
>
> **[Jain et al. 2014]**
> Prateek Jain, Ambuj Tewari, and Purushottam Kar. On iterative hard thresholding methods
> for high-dimensional m-estimation. Advances in neural information processing systems, 27,2014.
>
> **[Wang et al. 2024]**
> Zezhi Wang, Jin Zhu, Junxian Zhu, Borui Tang, Hongmei Lin, and Xueqin Wang. Sparsity-
> constraint optimization via splicing iteration. arXiv preprint arXiv:2406.12017, 2024.

---

> > ### Comment · Reviewer_PtcT · 2024-08-08
> >
> > Dear authors,
> >
> > Thank you so much for the detailed response and additional results. Most of my concerns have been cleared.
> >
> > I hope this discussion and clarification can be incorporated into the revised manuscript, which should benefit the paper's quality and contribution.
> >
> > I've increased my rating to 7.

---

> > > ### Author Response · Authors · 2024-08-09
> > >
> > > Thank you again for the helpful comments and critiques!
> > > We will add the discussions, clarifications, and additional numerical results to the revised manuscript.

---

### Official Review · Reviewer_vYuH · 2024-07-11

**Soundness:** 3
**Presentation:** 3
**Contribution:** 2
**Rating:** 7
**Confidence:** 3

**Summary:**

In this paper, the author has proposed a new method for solving the high-dimensional output regression (MOR) problem through compression. The authors introduce a two-stage framework that incorporates output compression to achieve computational efficiency while maintaining accuracy. Theoretical results demonstrate the framework's scalability and error bounds, while empirical results validate its performance.

**Strengths:**

- The author clearly defined the question about the MOR, and the proposed methods that trying to achieve the good performance.
- The proof and the structure of the paper is well-written and well-stated, containing comprehensive theoretical analyses, including training loss bounds and convergence guarantees.

**Weaknesses:**

- [Comparison Methods] In this paper, the authors has compare the model performance with some existing prediction baselines like Orthogonal Matching Pursuit and Elastic Net. Both of those two methods seems not to be the latest sparse regression methods that would be interesting to compare with. Therefore, I am wondering if some methods related to sparse high dimension regression like [1,2,3,4] can be discussed within the paper, and tested on the model performances. Also, there are plenty of the methods that are mentioned within the literature review parts, those methods are also not taken into considerations instead of elasticnet, which make the conclusion less convincing from the numerical results perspective.

- [Missing of the Sparsity levels] with the synthetic dataset. The paper doesn't mentioned the sparsity level while the sparsity level is pre-defined within the numerical experimental settings since the set up sparsity level tends to be the most correct baseline for evaluating the sparsity level of the model, so it would be really important to be shown within the paper.

[1] Bertsimas, D., and B. Van Parys. "Sparse high-dimensional regression: Exact scalable algorithms and phase transitions (2017)." arXiv preprint arXiv:1709.10029 (2019).
[2] Bertsimas, Dimitris, and Bart Van Parys. "SPARSE HIGH-DIMENSIONAL REGRESSION." The Annals of Statistics 48.1 (2020): 300-323.
[3] Liu, Liu, et al. "High dimensional robust sparse regression." International Conference on Artificial Intelligence and Statistics. PMLR, 2020.
[4] Sun, Qiang, et al. "Sprem: sparse projection regression model for high-dimensional linear regression." Journal of the American Statistical Association 110.509 (2015): 289-302.

**Questions:**

- [The influence on the compressed hyperparameter m] With the increase of the hyperparameter $m$ as the compressed dimension. The model performance of the methods has been significantly improved. Therefore, I am wondering if the model training time is depending on m, so it would be interesting to see how the training time increase in real datasets as the hyperparameter m change since the current training time is just a range, while the computational complexity O(KM) in the second stage, so the training time wshould definitely increase with the improvement of m.

- [The selection of m] I can understand there is a trade-off between model performance and compressed output dimension. However, if the dataset is totally unknown, the training time would increase under the trial. Therefore, I am wondering if that is also an issue when this method is finally applied for real application.

---

> ### Author Rebuttal · Authors · 2024-08-06
>
> ## Response
> We sincerely thank the reviewer for their positive assessment of the paper's clarity, methodology, and theoretical contributions. Figure1 mentioned below can be found in the uploaded pdf file in 'global' author rebuttal.
>
> **[Response to Comparison with Sparse Regression and related references]**
>
> Thank you for these suggestions on related literature and possible baselines. We will add the following comparisons and literature review to the revised version. Additionally, some numerical experiments will be conducted as suggested.
>
> **Proposed prediction stage vs. Sparse regression:**  Please refer to the 'global' author rebuttal part "Proposed prediction stage vs. Sparse regression" for detailed discussions.
>
> We will also add the following literature reviews (Bertsimas et al. 2019, Bertsimas et al. 2020, Liu et al. 2020, Sun et al. 2015) to Section 3 in our camera-ready version:
>
> - **[Bertsimas et al. 2019, Bertsimas et al. 2020]:** This work proposes a binary convex reformulation (BCR) of the sparse regression with an additional $\ell_2$ squared regularizer and devises a cutting plane method for solving the proposed BCR. The authors of this work demonstrate its scalability under their sample-generating model. Compared with our result, the proposed cutting-plane method lacks an iterative convergence guarantee, even though such a method ensures the finding of a globally optimal solution with exponential time complexity.
>
> - **[Liu et al. 2020]:** This paper proposes an algorithm for high-dimensional sparse regression with a constant fraction of corruptions, which has a different problem setting compared with ours.
>
> - **[Sun et al. 2015]:** This paper develops a sparse projection regression modeling (SPReM) framework to perform multivariate regression, focusing on addressing the low statistical power issue of many standard statistical approaches. However, the authors of Sun et al. 2015 do not provide any iterative convergence result for their proposed coordinate descent algorithm (Algorithm 1).
>
>
>
> **[Response to Missing of Sparsity levels]**
>
> Thank you for your feedback regarding the importance of including the sparsity levels in the discussion of our synthetic dataset experiments. We would like to point out that the sparsity levels are indeed mentioned on line 298 on page 8, where $s = 3$. However, we understand that this might not have been sufficiently emphasized. To address this, we will make the description more explicit and highlight it in figure captions and subplot titles.
>
>
>
> **[Response to the influence on the compressed output dimension $m$: model performance]**
>
> Here, the performances are relatively stable as $m$ is greater than some worst-case lower bound. Typically, the worst-case lower bound is a fixed number much smaller than the label number $K$ given the training dataset (see response on "The selection of $m$" below). In our numerical experiment, due to resource limitations, we only conducted several experiments on relatively small real datasets where $K \approx 31,000$. Thus, the ratio of the worst-case lower bound on compression number divided by label numbers $K$ is high compared to large datasets, leading to the result that the accuracy of experiments significantly improves as $m$ increases. Now, we have tested on relatively larger $m$; see the third panel in Figure 1 and the Real-world dataset part in the global author rebuttal for detailed discussions.
>
> **[Response to the influence on the compressed output dimension $m$: computational complexity]**
>
> Thank you for pointing this out. Yes, the computational complexity will increase with the improvement of $m$. As noted above, the accuracy performance is relatively stable as $m$ is greater than some worst-case lower bound. Thus, there is no strong motivation to increase hyperparameter $m$ beyond such a worst-case lower bound, which leads to an "upper bound" on the computational complexity. See the second panel in Figure 1 in the global rebuttal for details.
>
> **[Response to the selection of $m$]**
>
> Thank you for your feedback regarding the selection of $m$ for real applications without prior knowledge. As noted in Remark 3 (lines 182-184), we provide a theoretical worst-case lower bound on $m$ with high probability ensuring that the compressed matrix $\Phi$ satisfies the RIP-property. More specifically, the worst-case bound is given by $m \ge \dfrac{12}{\delta^2}(3s \ln{9} + 3s\ln{\dfrac{e|\mathcal{V}|}{3s}}+\ln{\dfrac{2}{\tau}})$ where $s$ represents the sparsity level. As implemented in our numerical experiments, we use the worst-case bound above and repeat the experiments ten times to reduce the impact of randomness. In real applications, the dimension of the dataset is typically known, and the sparsity level $s$ can be chosen close to the average of non-zero labels in the dataset, allowing us to compute the worst-case lower bound of $m$​ accordingly.
>
>
>
> ## Reference
>
> **[Bertsimas et al. 2019]**
> D Bertsimas and B Van Parys. Sparse high-dimensional regression: Exact scalable algorithms
> and phase transitions (2017). arXiv preprint arXiv:1709.10029, 2019.
>
> **[Bertsimas et al. 2020]**
> Dimitris Bertsimas and Bart Van Parys. Sparse high-dimensional regression. The Annals of
> Statistics, 48(1):300–323, 2020.
>
> **[Liu et al. 2020]**
> Liu Liu, Yanyao Shen, Tianyang Li, and Constantine Caramanis. High dimensional robust
> sparse regression. In International Conference on Artificial Intelligence and Statistics, pages
> 411–421. PMLR, 2020.
>
> **[Sun et al. 2015]**
> Qiang Sun, Hongtu Zhu, Yufeng Liu, and Joseph G Ibrahim. Sprem: sparse projection re-
> gression model for high-dimensional linear regression. Journal of the American Statistical
> Association, 110(509):289–302, 2015.

---

> > ### Comment · Reviewer_vYuH · 2024-08-08
> >
> > Thank you for your response, and that solves my concerns to the paper. I have slightly increase my rating for this paper.

---

> > > ### Author Response · Authors · 2024-08-09
> > >
> > > Thank you for the insightful comments and willingness to raise your score! We will add the related references, discussions, clarifications, and additional numerical results to the revised manuscript.

---

### Official Review · Reviewer_hQ2t · 2024-07-11

**Soundness:** 4
**Presentation:** 3
**Contribution:** 3
**Rating:** 6
**Confidence:** 2

**Summary:**

This manuscript proposes a new approach the authors refer to as Sparse & High-dimensional-Output REgression (SHORE) to tackle linear regression problems promoting sparse predictions. A major component of the author's approach is to reduce the computational cost with a random normal matrix compressing the signal first and then producing sparse predictions in the second phase. The manuscript provides a theoretical analysis of the accuracy of the sparse prediction. The authors include further some numerical numerical investigations supporting their theoretical findings.

**Strengths:**

The manuscript is well-written and tackles an interesting problem.

**Weaknesses:**

Comparison to state-of-the-art methods such as FISTA (Fast Iterative Shrinkage-Thresholding Algorithm) or ADMM (Alternating Direction Method of Multipliers) would strengthen the contribution of this work. A clear comparison highlighting the advantages of the proposed method in terms of convergence speed, memory efficiency, or specific problem suitability would be valuable. Additionally, the paper could benefit from exploring connections to randomized linear algebra, particularly regarding the selection of the matrix Φ. The paper Randomized Numerical Linear Algebra: Foundations & Algorithms by Martinsson and Tropp provides a comprehensive overview of how random projections can be leveraged to achieve efficient solutions in high-dimensional problems. Further, Equation (1) is a simple linear regression or least squares problem  ||Y-ZX||_F^2 =  ||\vec(Y) - (X'  \kron  I) vec(Z)||_2^2 which does not require a new renaming as SHORE.

**Questions:**

See weaknesses.

**Limitations:**

yes.

---

> ### Author Rebuttal · Authors · 2024-08-06
>
> ## Response
> Thank you for your positive feedback! Figure1 mentioned below can be found in the uploaded pdf file in 'global' author rebuttal.
>
> **[Response to FISTA or ADMM]** Thank you for your comments. We have conducted some preliminary numerical experiments concerning comparison with FISTA, as you suggested. Please check the first panel in Figure 1 in the global response. Besides, it is difficult for FISTA to achieve the stopping criteria in Appendix 6.1 on page 21. We suspect that this is due to the special structure of our proposed model versus the general SCO.
>
>
> **[Response to other randomized linear algebra]** Thank you for providing these references. Actually, as you mentioned above, we realized that how to select a projection matrix is an important open problem that may significantly influence the accuracy of the final predicted output in both theory and practice (also see Blum et al. (2005), Zhang et al. (2015), Pillai et al. (2011)). We will add this part to Section 5 as one possible future direction.
>
>
> **[Response to SHORE name issue]** Thank you for raising this confusion. We will add a discussion to the camera-ready version based on the following explanations. We agree that Equation (1) for the training stage can be reformulated as a linear regression, as you presented. However, the SHORE framework we proposed here refers to a two-stage framework, which includes both the training stage (Equation (2), compressed version of Equation (1)) and the prediction stage (Equation (3)). Moreover, we would like to mention that vectorizing the output matrix $\boldsymbol{Y}$ and linear regressor $\boldsymbol{Z}$ might impede its ability to extend to other structured settings. For example, in some specific applications, the linear regressor $\boldsymbol{Z}$ is assumed to be low-rank, which is more natural to formulate $\boldsymbol{Z}$ as a matrix rather than a vector.
>
> ## Reference
> **[Blum et al. (2005)]**
> Avrim Blum. Random projection, margins, kernels, and feature-selection. In International Statistical and Optimization Perspectives Workshop” Subspace, Latent Structure and Feature Selection”, pages 52–68. Springer, 2005.
>
> **[Zhang et al. (2015)]**
> Shengping Zhang, Huiyu Zhou, Feng Jiang, and Xuelong Li. Robust visual tracking using
> structurally random projection and weighted least squares. IEEE Transactions on Circuits
> and Systems for Video Technology, 25(11):1749–1760, 2015.
>
> **[Pillai et al. (2011)]**
> Jaishanker K Pillai, Vishal M Patel, Rama Chellappa, and Nalini K Ratha. Secure and robust
> iris recognition using random projections and sparse representations. IEEE transactions on
> pattern analysis and machine intelligence, 33(9):1877–1893, 2011.

---

> > ### Comment · Reviewer_hQ2t · 2024-08-12
> >
> > Thank you for addressing my concerns. I have slightly increased my rating.

---

> > > ### Author Response · Authors · 2024-08-13
> > >
> > > Thank you for your response. We truly appreciate your feedback and would be grateful if you could consider raising your rating score, as we have taken steps to address the concerns you mentioned.

---

### Author Rebuttal · Authors · 2024-08-06

## Responses to All Reviewers & Area Chairs

We would like to thank the reviewers for their constructive and high-quality feedback. The manuscript has been revised based on the comments given in three reviewers' reports.

### Comparisons & Differences

We first highlight the differences between our proposed prediction stage, general sparsity-constrained optimization (SCO), sparse regression, and the classical manner (i.e., prediction stage of SHORE without compression). The following discussions will be added to the revised version.

- **Proposed prediction stage vs. General SCO:** The prediction stage requests solving a sparsity-constrained optimization (SCO) problem with its general form: $\min_{||\boldsymbol{\alpha}||_0 \leq k} ||\boldsymbol{\beta} - \boldsymbol{A}\boldsymbol{\alpha}||_2^2$. Finding the global optimal solution of SCO, in general (worst-case scenario), is computationally hard. In contrast with the general SCO, the problem we studied in the predicted stage takes the random projection matrix $\boldsymbol{\Phi}$ with restricted isometry property (RIP) as its $\boldsymbol{A}$ and uses $\widehat{\boldsymbol{W}}\boldsymbol{x}$ with $\widehat{\boldsymbol{W}}$ obtained from the compressed training stage as its $\boldsymbol{\beta}$. As a result, and as presented in Theorem 2 (page 5) and Theorem 4 (page 7), the proposed efficient algorithm (Algorithm 1 -- PGD) ensures a globally linear convergence to a ball with center $\widehat{\boldsymbol{y}}$ (optimal solution of the prediction stage) and radius $O(||\boldsymbol{\Phi}\widehat{\boldsymbol{y}} - \widehat{\boldsymbol{W}}\boldsymbol{x}||_2)$, which might not be true for general SCO problems.

- **Proposed prediction stage vs. Sparse regression:** Although both the proposed prediction stage and sparse high-dimensional regression share a similar optimization formulation:
  $\min_{||\boldsymbol{\beta}||_0 \leq k} ~~ ||\boldsymbol{y} - \boldsymbol{X}^{\top}\boldsymbol{\beta}||_2^2,$
  the proposed prediction stage (equation (3) under line 149 on page 4) is distinct from sparse regression in the following parts:

  - **Underlying Model:** Most existing works on sparse high-dimensional regression assume that samples are i.i.d. generated from the linear relationship $\boldsymbol{y} = \boldsymbol{X}^{\top}\boldsymbol{\beta}^* + \boldsymbol{\epsilon}$ with underlying sparse ground truth $\boldsymbol{\beta}^*$. In the proposed prediction stage, we do not assume additional underlying models on samples if there is no further specific assumption. The problem we studied in the prediction stage takes the random projection matrix $\boldsymbol{\Phi}$ with RIP as its $\boldsymbol{X}^\top$ (whereas $\boldsymbol{X}^{\top}$ in sparse regression does not ensure RIP) and uses $\widehat{\boldsymbol{W}}\boldsymbol{x}$ with $\widehat{\boldsymbol{W}}$ obtained from the compressed training stage as its $\boldsymbol{Y}$.
  - **Problem Task:** Sparse regression aims to recover the sparse ground truth $\boldsymbol{\beta}^*$ given a sample set $\{(\boldsymbol{x}^i, \boldsymbol{y}^i)\}_{i = 1}^n$ with $n$ i.i.d. samples. In contrast, the task of the proposed prediction stage is to predict a sparse high-dimensional output $\widehat{\boldsymbol{y}}$ given a random projection matrix $\boldsymbol{\Phi}$ and a single input $\boldsymbol{x}$.

  Therefore, due to distinct underlying models and problem tasks, some existing methods/results for sparse regression cannot be directly applied to the proposed prediction stage.
- **Proposed prediction stage vs. Classical manner:** As mentioned in Remark 1 & Remark 2 on page 4, the compressed SHORE (both training stage & prediction stage) enjoys better computational complexity with respect to the classical manner, especially under the setting with high-dimensional outputs.

### Numerical Experiments
We then report some preliminary results of numerical experiments as suggested by reviewers (see Figure 1 and Table 1). Other numerical experiments, including relatively sophisticated optimization methods or large real instances, will be reported later in the camera-ready version due to resource and time limitations.
- **Comparing with FISTA:** The first panel in Figure 1 compares the proposed algorithm (Algorithm 1 -- PGD) with FISTA. As we can observe, the precision of the proposed algorithm outperforms the precision of FISTA, especially when $m$ is small (heavily compressed).
- **Prediction time:** The second panel in Figure 1 reports the prediction running time (measured in seconds) on solving one implemented prediction method with early stopping (see lines 618-620, Section A.6.1 on page 21) by PGD under different compressed output dimensions $m$. As we can observe, the running time first decreases dramatically, then increases almost linearly with respect to $m$. Such a phenomenon has occurred since the max number of iterations $T$ is 30 in the implemented prediction method with early stopping, which is relatively large; As $m$ increases but is still less than 1000, the actual number of iterations drops dramatically due to early stopping criteria; After $m$ passes 1000, the actual number of iterations stays at 5, and then the running time grows linearly as dimension $m$ increases.
- **Real-world dataset:** The third panel in Figure 1 shows that for a real-world dataset with 31,000 labels, the model performance of our proposed algorithm becomes stable as $m$ is greater than some worst-case lower bound.
- **Comparing with elastic net under low SNR regime:** As observed from Table 1, the elastic net has similar performances on precision compared with the performances of the proposed PGD under a low SNR regime. It's true that the elastic net performs much better in numerical experiments in low SNR regime than in high SNR regime. We would like to point out that the elastic net method performs much slower than PGD. The prediction time for the elastic net is around 1,000 seconds, while the time for the PGD is around 10 seconds.

---

### Decision · Program_Chairs · 2024-09-25

**Decision:**

Accept (poster)

**Comment:**

The authors propose in this article a two-stage approach to tackle multi-output regression under a pre-determined sparsity level on the output vector and provide performance guarantees for this computational efficient method. Several points raised by the reviewers regarding the comparison to other methods, the experimental study and the clarity were effectively addressed by the authors' rebuttal, leading to a consensus to accept the paper, with recommendation to incorporate the discussion and the additional results into the final version.